EMBO
Molecular Medicine

# The role of interleukin-6 in monitoring severe case of coronavirus disease 2019

Tao Liu[1,†] , Jieying Zhang[1,†], Yuhui Yang[1,†], Hong Ma[1], Zhenyu Li[1], Jiaoyue Zhang[2], Ji Cheng[3], Xiaoyun Zhang[4], Yanxia Zhao[1], Zihan Xia[1], Liling Zhang[1,*] , Gang Wu[1,**] & Jianhua Yi[5,***]

## Abstract

Progression to severe disease is a difficult problem in treating coronavirus disease 2019 (COVID-19). The purpose of this study is to explore changes in markers of severe disease in COVID-19 patients. Sixty-nine severe COVID-19 patients were included. Patients with severe disease showed significant lymphocytopenia. Elevated level of lactate dehydrogenase (LDH), C-reactive protein (CRP), ferritin, and D-dimer was found in most severe cases. Baseline interleukin-6 (IL-6) was found to be associated with COVID-19 severity. Indeed, the significant increase of baseline IL-6 was positively correlated with the maximal body temperature during hospitalization and with the increased baseline of CRP, LDH, ferritin, and D-dimer. High baseline IL-6 was also associated with more progressed chest computed tomography (CT) findings. Significant decrease in IL-6 and improved CT assessment was found in patients during recovery, while IL-6 was further increased in exacerbated patients. Collectively, our results suggest that the dynamic change in IL-6 can be used as a marker for disease monitoring in patients with severe COVID-19.

**Keywords** biomarker; coronavirus disease 2019; cytokine storm; disease monitoring; interleukin-6

**Subject Categories** Immunology; Microbiology, Virology & Host Pathogen Interaction

## Introduction

In early December 2019, the first cases of pneumonia of unknown origin were reported in Wuhan City, Hubei Province, China. The pathogen has been identified as a novel β-coronavirus by full-genome sequencing and is named severe acute respiratory syndrome coronavirus 2 (SARS-CoV-2) which shares phylogenetic similarity with severe acute respiratory syndrome coronavirus (SARS-CoV) that caused the outbreak of SARS in 2003 (Luk *et al*, 2019; Guan *et al*, 2020; Zhou *et al*, 2020a). Disease caused by SARS-CoV-2, which has been highly contagious and spread rapidly nationwide and worldwide, has been designated coronavirus disease 2019 (COVID-19) by World Health Organization (WHO). The epidemic curves of COVID-19 reflected what might be a mixed outbreak pattern, with the early cases suggested a continuous common source, while the later cases suggested a propagated source as the virus began to be transmitted from person to person (Wu & McGoogan, 2020). Most cases are mild to moderate and curable, and the overall crude mortality rate is low (Guan *et al*, 2020). However, a proportion of patients with severe disease characterized by respiratory dysfunction have shown a high mortality rate (Guan *et al*, 2020; Yang *et al*, 2020). By February 16, 2020, a total number of 57,934 cases have been reported in mainland China, including 10,644 severe cases. As the outbreak area of COVID-19, the total number of confirmed cases in Hubei is 49,847, with 9,797 (19.6%) severe cases and an overall mortality rate of 3.4%. The disease incidence and mortality rate of severe COVID-19 in Hubei is relatively high as compared with other areas in China.

At present, due to the lack of reliable marker and effective antiviral medication, the monitoring of severe cases of COVID-19 mainly relies on the observation of clinical presentation (Wang *et al*, 2020; Yang *et al*, 2020). In infections caused by highly pathogenic coronavirus, such as SARS and MERS, previous studies have suggested that lymphocytopenia and inflammatory cytokine storm are typical abnormalities and are considered disease severity related (Tanaka *et al*, 2014; de Brito *et al*, 2016; Rose-John *et al*, 2017; Gupta *et al*, 2020). Similarly, a decrease in lymphocyte count and an increase in inflammatory cytokines in peripheral blood have been reported in COVID-19 patients (Magnan *et al*, 1996; Huang *et al*, 2020; Liu *et al*, 2020a). Given the high mortality rate of severe COVID-19 cases, a better understanding of the clinical features is urgently needed and may help screen out reliable

1   Cancer Center, Union Hospital, Tongji Medical College, Huazhong University of Science and Technology, Wuhan, China
2   Department of Endocrinology, Union Hospital, Tongji Medical College, Huazhong University of Science and Technology, Wuhan, China
3   Department of Gastrointestinal Surgery, Union Hospital, Tongji Medical College, Huazhong University of Science and Technology, Wuhan, China
4   Liver intensive care unit, Zhongshan Hospital, Fudan University, Shanghai, China
5   Department of Infectious Diseases, Union Hospital, Tongji Medical College, Huazhong University of Science and Technology, Wuhan, China
   *Corresponding author. Tel: +86 027 85873100; E-mail: lily-1228@hotmail.com
   **Corresponding author. Tel: +86 027 85873100; E-mail: wuganghustxh@163.com
   ***Corresponding author. Tel: +86 027 85873100; E-mail: doctor_yi2017@163.com
   †These authors contributed equally to this work

markers for inflammation monitoring through the course of disease. According to previous relevant literature on viral pneumonia and the current therapeutic experience on severe type COVID-19, the cytokine storm may be the main reason for rapid disease progression and poor treatment response (Yiu et al, 2012; de Brito et al, 2016; Gupta et al, 2020). In this study, by examining the IL-6 and parameters relevant to other systemic inflammatory response and clinical severity, such as body temperature, CRP level, erythrocyte sedimentation rate (ESR), and chest CT findings, we analyzed the clinical characteristics and inflammatory markers in patients with severe type COVID-19 in Wuhan City to explore potential markers for disease monitoring.

## Results

### Demographic and clinical characteristics

A total of 69 severe cases were included. The median age was 56 years. Female accounted for 52.17% of the enrolled cases. The smoking rate was 11.59%. Fever (79.72%), cough (63.77%), shortness of breath (57.97%), and fatigue (50.72%) were the most common symptoms, while diarrhea (15.94%), vomiting (7.25%), and sore throat (5.80%) were relatively less common. 36.23% of patients had at least one comorbidity (e.g., hypertension, chronic obstructive pulmonary disease, coronary heart disease, and hepatitis B virus infection), and there were four cases of tumor. Another 11 non-severe cases were also enrolled (Table 1).

### Radiologic and laboratory findings

Table 2 shows the radiologic and laboratory findings on admission. The most common CT patterns in severe cases upon admission were bilateral patchy shadowing (60.87%) and interstitial abnormalities (27.54%), while the common type mainly manifested as focal ground glass opacity and patchy shadowing (54.5%; Table 2).

Baseline neutrocytopenia, lymphocytopenia, and thrombocytopenia were observed in 13.04, 79.71, and 24.64% of the severe type patients, respectively. Lymphocyte count in severe case was significantly less than the non-severe cases (Fig 1A). Increased creatine kinase (CK) was present in 14.49% of the cases. Elevated level of alanine aminotransferase and aspartate aminotransferase was more common, and both were detected in 37.68% of the cases (Table 2 and Fig 1B).

For inflammatory parameters, abnormal baseline PCT was not common in both non-severe and severe patients. However, patients with severe disease showed significantly elevated level of D-dimer, ESR, LDH, CRP, and ferritin than those with non-severe disease (Fig 1C–E).

### Treatment and clinical outcomes

89.86% of the severe type patients received antibiotics, 63.77% received antiviral therapy (20.29% with oseltamivir, 52.17% with umifenovir, and 7.25% with lopinavir/ritonavir), 42.03% received glucocorticoids, and 50.72% received human immunoglobulin

(Table 3). 55.07% of the severe cases needed oxygen therapy, among which 27.54% received high-flow oxygen therapy, non-invasive ventilation, or invasive ventilation (Table 3).

Among the enrolled patients with severe disease, 53.62% were recovered and discharged, 46.38% were still hospitalized, three patients (4.35%) needed transfer to the intensive care unit, and no death case occurred (Table 3). Seven patients developed acute respiratory distress syndrome (ARDS), and one got septic shock (Table 3). Compared with patients with non-severe disease, the time from symptom onset to initial COVID-19 diagnosis and to development of pneumonia in patients with severe disease was shorter, but it was not significant ($P > 0.05$; Fig 2A and B). There was no significant difference in the time from symptom onset to treatment, and the time from development of pneumonia to recovery in patients with severe disease was longer ($P > 0.05$; Fig 2C and D).

### Immunological findings

The baseline proportion of $CD4^+$ T cells, $CD8^+$ T cells, B cells, natural killer (NK) cells, and the $CD4^+$ T cells/$CD8^+$ T cells ratio are basically within the normal range (Fig 1F–H). The baseline level of IL-2, IL-4, IL-10, TNF-$\alpha$, and IFN-$\gamma$ was within normal range, while IL-10 was slightly increased (Fig 1I–K).

The IL-6 level was increased significantly on admission in severe cases as compared with the non-severe cases (Fig 1D). The baseline elevated IL-6 in severe COVID-19 cases was positively correlated with the maximal body temperature during hospitalization ($r = 0.521$, $P < 0.001$; Fig 3A). Meanwhile, it was positively correlated with the increase of CRP ($r = 0.781$, $P < 0.001$), LDH ($r = 0.749$, $P < 0.001$), ferritin ($r = 0.606$, $P < 0.001$), and D-dimer ($r = 0.679$, $P < 0.001$; Fig 3B–E). Severe COVID-19 patients who received high-flow oxygen inhalation and mechanical ventilation during hospitalization showed significant higher baseline IL-6 level than those who did not (Fig 3F). Patients who received administration of glucocorticoids also showed a tendency of higher baseline IL-6 level, though the difference was not significant.

For the 30 patients whose IL-6 was assessed before and after treatment, 26 patients, classified into three groups according to the baseline IL-6 level of $\geq 40$, $\geq 20$, or $< 20$ pg/ml, showed significantly reduced IL-6 after treatment (Fig 4A, left). Higher baseline IL-6 was associated with more severe pulmonary damage as evidenced by chest CT scans, and the decrease in IL-6 after treatment was accompanied with improved CT assessments after treatment (Fig 4A, right). Disease of four patients exacerbated progressively. Among them, three patients showed increased IL-6 as compared with the baseline (Fig 4B), while the change in IL-6 in one patient was not obvious. For this patient, there was an increase in PCT level, and the progression of lung disease was confirmed as bacterial, but not viral, pneumonia, as evidenced by sputum culture positive for bacteria growth (Fig 4C).

In another patient who presented with disease aggravation and subsequent alleviation evidenced by CT scans (Fig 5A), the IL-6 and CRP varied correspondingly, and more importantly, the occurrence of abnormal IL-6 and CRP was prior to the changes detected by CT assessment (Fig 5D and E). The procalcitonin (PCT) level had stayed within the normal range throughout the course of the

**Table 1. Clinical characteristics of patients with severe COVID-19.**

| Characteristics | All patients (*N* = 80) | Non-severe (*N* = 11) | Severe (*N* = 69) | *P* value |
|---|---|---|---|---|
| Age—median year (range) | 53.00 (26.00–86.00) | 31.00 (26.00–58.00) | 56.00 (27.00–86.00) | < 0.001 |
| Gender—no. (%) | | | | |
| Male | 34 (42.50) | 1 (9.09) | 33 (47.83) | < 0.037 |
| Female | 46 (57.50) | 10 (90.91) | 36 (52.17) | |
| Smoking history—no. (%) | | | | |
| Non-smoker | 72 (90.00) | 11 (100.00) | 61 (88.41) | 0.591 |
| Smoker | 8 (10.00) | 0 (0.00) | 8 (11.59) | |
| Fever on admission—no. (%) | 64 (80.00) | 9 (81.82) | 55 (79.72) | 0.808 |
| Temperature—no. (%) | | | | |
| < 37.5°C | 17 (21.25) | 2 (18.18) | 15 (21.74) | |
| 37.5–38.0°C | 19 (23.75) | 4 (36.36) | 15 (21.74) | |
| 38.1–39.0°C | 38 (47.50) | 5 (45.45) | 33 (47.83) | |
| > 39.0°C | 6 (7.50) | 0 (0.00) | 6 (8.70) | |
| Fever during hospitalization—no. (%) | 46 (57.50) | 2 (18.18) | 44 (63.77) | |
| Highest temperature during hospitalization—no. (%) | | | | |
| < 37.5°C | 35 (43.75) | 10 (90.91) | 25 (36.23) | 0.012 |
| 37.5–38.0°C | 16 (20.00) | 1 (9.09) | 15 (21.74) | |
| 38.1–39.0°C | 23 (28.75) | 0 (0.00) | 23 (33.33) | |
| > 39.0°C | 6 (7.50) | 0 (0.00) | 6 (8.70) | |
| Symptoms—no. (%) | | | | |
| Conjunctival congestion | 1 (1.25) | 0 (0.00) | 1 (1.45) | 1.000 |
| Nasal congestion | 5 (6.25) | 0 (0.00) | 5 (7.25) | 1.000 |
| Headache | 8 (10.00) | 0 (0.00) | 8 (11.59) | 0.516 |
| Cough | 53 (66.25) | 9 (81.82) | 44 (63.77) | 0.405 |
| Sore throat | 8 (10.00) | 4 (36.36) | 4 (5.80) | 0.009 |
| Sputum production | 15 (18.75) | 1 (9.09) | 14 (20.29) | 0.640 |
| Fatigue | 36 (45.00) | 1 (9.09) | 35 (50.72) | 0.024 |
| Hemoptysis | 2 (2.50) | 2 (18.18) | 0 (0.00) | 0.017 |
| Shortness of breath | 44 (55.00) | 4 (36.36) | 40 (57.97) | 0.312 |
| Nausea or vomiting | 7 (8.75) | 2 (18.18) | 5 (7.25) | 0.245 |
| Diarrhea | 15 (18.75) | 4 (36.36) | 11 (15.94) | 0.232 |
| Myalgia or arthralgia | 12 (15.00) | 0 (0.00) | 12 (17.39) | 0.296 |
| Shivering | 13 (16.25) | 0 (0.00) | 13 (18.84) | 0.257 |
| Physical signs—no. (%) | | | | |
| Throat congestion | 3 (3.75) | 0 (0.00) | 3 (4.35) | 1.000 |
| Tonsil swelling | 0 (0.00) | 0 (0.00) | 0 (0.00) | |
| Lymph node enlargement | 0 (0.00) | 0 (0.00) | 0 (0.00) | |
| Skin rash | 0 (0.00) | 0 (0.00) | 0 (0.00) | |
| Coexisting disorders—no. (%) | | | | |
| Any of the following | 28 (35.00) | 3 (27.27) | 25 (36.23) | 0.812 |
| Diabetes | 11 (13.75) | 0 (0.00) | 11 (15.94) | 0.340 |
| Hypertension | 14 (17.50) | 0 (0.00) | 14 (20.29) | 0.223 |
| Coronary heart disease | 6 (7.50) | 0 (0.00) | 6 (8.70) | 0.589 |
| Cerebrovascular diseases | 0 (0.00) | 0 (0.00) | 0 (0.00) | |

**Table 1.** (continued)

| Characteristics | All patients (*N* = 80) | Non-severe (*N* = 11) | Severe (*N* = 69) | *P* value |
|---|---|---|---|---|
| Hepatitis B virus infection | 1 (1.25) | 0 (0.00) | 1 (1.45) | 1.000 |
| Cancer[a] | 7 (8.75) | 3 (27.27) | 4 (5.80) | 0.051 |
| Chronic renal diseases | 0 (0.00) | 0 (0.00) | 0 (0.00) | |
| Immunodeficiency | 0 (0.00) | 0 (0.00) | 0 (0.00) | |

*P* values denoted the comparison between non-severe and severe cases.
[a]Cancer referred to any type of malignancy. All cases were stable disease.

**Table 2.** Radiologic and laboratory findings of patients with severe COVID-19.

| Radiologic and laboratory findings | All patients (*N* = 80) | Non-severe (*N* = 11) | Severe (*N* = 69) | *P* value |
|---|---|---|---|---|
| **Radiologic findings** | | | | |
| Abnormalities on chest CT—no. (%) | | | | |
| Ground glass opacity | 3 (2.50) | 3 (27.27) | 0 (0.00) | 0.017 |
| Local patchy shadowing | 5 (6.25) | 3 (27.27) | 2 (2.90) | 0.017 |
| Bilateral patchy shadowing | 47 (58.75) | 5 (45.45) | 42 (60.87) | 0.526 |
| Interstitial abnormalities | 19 (23.75) | 0 (0.00) | 19 (27.54) | 0.107 |
| **Blood cell count—no. (%)** | | | | |
| Neutrophil count | | | | |
| > $6.3*10^9$/l | 9 (11.25) | 0 (0.00) | 9 (13.04) | |
| $1.8$–$6.3*10^9$/l | 62 (77.50) | 11 (100.00) | 51 (73.91) | |
| < $1.8*10^9$/l | 9 (11.25) | 0 (0.00) | 9 (13.04) | |
| Lymphocyte count | | | | |
| < $1.5*10^9$/l | 60 (75.00) | 5 (45.45) | 55 (79.71) | 0.039 |
| Mean ± SD | 1.11 ± 0.51 | 1.61 ± 0.39 | 1.03 ± 0.48 | < 0.001 |
| Platelet count | | | | |
| < $150*10^9$/l | 17 (21.25) | 0 (0.00) | 17 (24.64) | 0.145 |
| Hemoglobin level (g/dl)—mean ± SD | 127.10 ± 15.32 | 127.8 ± 16.13 | 122.7 ± 7.82 | 0.314 |
| **Distribution of other findings—no. (%)** | | | | |
| C-reactive protein level ≥ 10 mg/l | 60 (75.00) | 1 (9.09) | 59 (85.51) | < 0.001 |
| Procalcitonin ≥ 0.5 ng/ml | 2 (2.50) | 0 (0.00) | 2 (2.90) | 1.000 |
| Lactose dehydrogenase ≥ 250 U/l | 46 (57.50) | 1 (9.09) | 45 (65.22) | 0.002 |
| Aspartate aminotransferase > 40 U/l | 27 (33.75) | 1 (9.09) | 26 (37.68) | 0.129 |
| Alanine aminotransferase > 40 U/l | 27 (33.75) | 1 (9.09) | 26 (37.68) | 0.129 |
| Total bilirubin > 17.1 μmol/l | 7 (8.75) | 1 (9.09) | 6 (8.70) | 1.000 |
| Creatine kinase ≥ 200 U/l | 10 (12.50) | 0 (0.00) | 10 (14.49) | 0.390 |
| Creatinine ≥ 133 μmol/l | 1 (1.25) | 0 (0.00) | 1 (1.45) | 1.000 |
| ᴅ-dimer ≥ 0.5 mg/l | 45 (56.25) | 0 (0.00) | 45 (65.22) | < 0.001 |
| Erythrocyte sedimentation rate (mm/h)—mean ± SD | 40.59 ± 27.2 | 19.91 ± 23.74 | 44.58 ± 26.16 | 0.001 |
| Ferritin (μg/L)—mean ± SD | 690.20 ± 864.3 | 155.70 ± 187.3 | 827.2 ± 916.9 | 0.001 |

Lymphocytopenia was defined as the lymphocyte count of < 1,500 per cubic millimeter. Thrombocytopenia was defined as the platelet count of < 150,000 per cubic millimeter.
*P* values denoted the comparison between non-severe and severe cases.
SD, standard deviation.

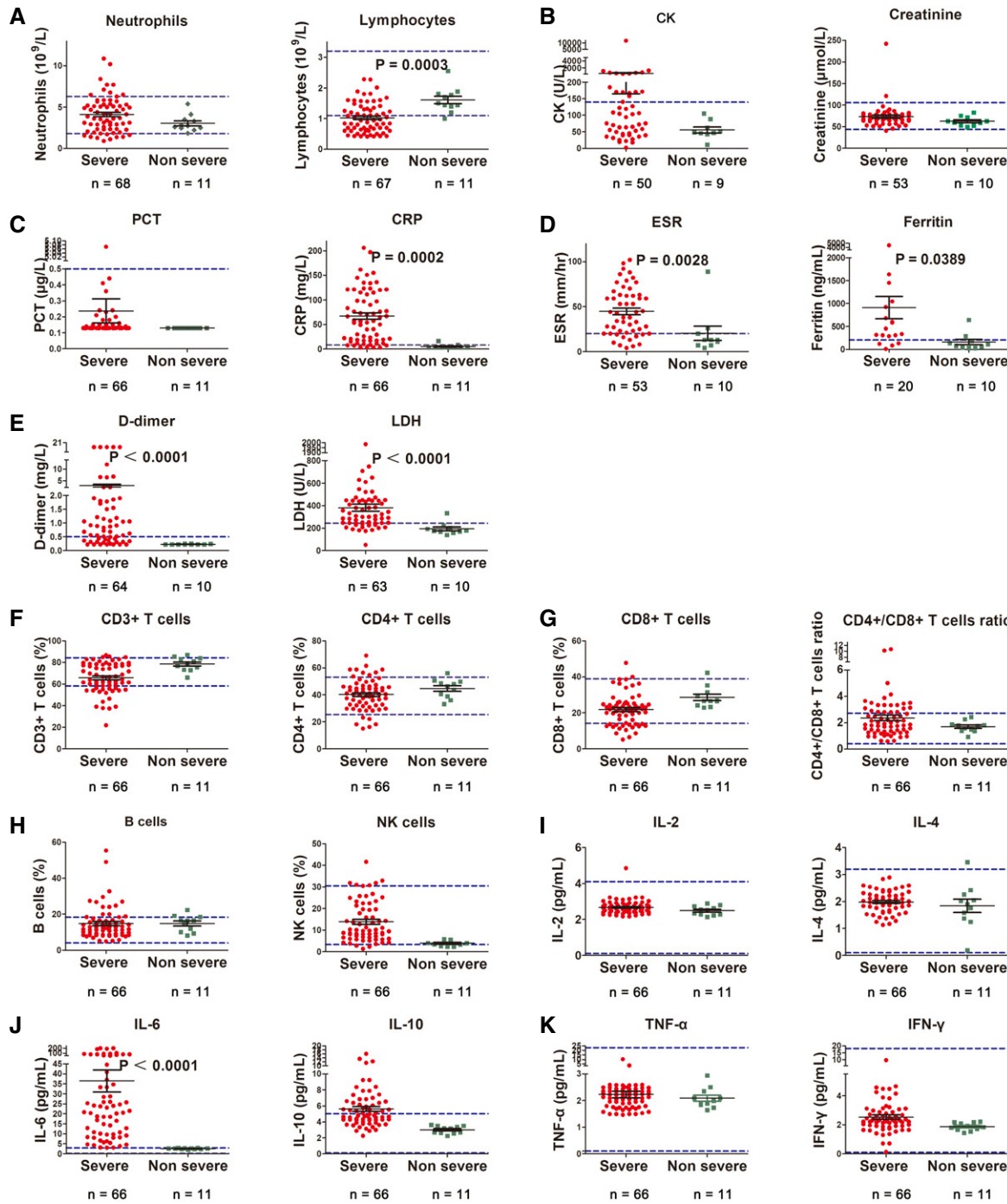

**Figure 1. Laboratory findings in severe versus non-severe COVID-19 patients.**

A Neutrophils and lymphocytes count.

B Levels of creatine kinase (CK) and creatinine.

C Levels of procalcitonin (PCT) and C-reactive protein (CRP).

D Levels of erythrocyte sedimentation rate (ESR) and ferritin.

E Levels of D-dimer and lactate dehydrogenase (LDH).

F Proportion of total CD3+ and CD4+ T cells.

G Proportion of CD8+ T cells and CD4+/CD8+ T cell ratio.

H Proportion of B and natural killer (NK) cells.

I–K Cytokine profile of COVID-19 patients for IL-2, IL-4 (I), IL-6, IL-10 (J), and TNF-α and IFN-γ (K).

Data information: Statistical analysis was performed by paired two-tailed Mann–Whitney U-test (A–D). Blue dotted lines denote normal value or normal range. Error bars, SEM.

disease, while the lymphocyte count began to decline 5 days after symptom onset and then returned to normal range (Fig 5B and C). The body temperature also correlated with the variation in IL-6 and CRP, as the body temperature stayed abnormal during the rising phase while returned to normal during the decline phase of IL-6 and CRP (Fig 5F).

Table 3. Complications, treatment, and clinical outcomes of patients with severe COVID-19.

| Characteristics | All patients (*N* = 80) | Non-severe (*N* = 11) | Severe (*N* = 69) | *P* value |
|---|---|---|---|---|
| **Complications—no. (%)** | | | | |
| Septic shock | 1 (1.25) | 0 (0.00) | 1 (1.45) | |
| Acute respiratory distress syndrome | 7 (8.75) | 0 (0.00) | 7 (10.14) | |
| Acute kidney injury | 0 (0.00) | 0 (0.00) | 0 (0.00) | |
| Disseminated intravascular coagulation | 0 (0.00) | 0 (0.00) | 0 (0.00) | |
| Rhabdomyolysis | 0 (0.00) | 0 (0.00) | 0 (0.00) | |
| **Time to different type of events during disease course—days** | | | | |
| From symptom onset to initial treatment | | | | |
| Median (interquartile range) | 1.00 (1.00–4.00) | 1.00 (1.00–5.00) | 1.00 (1.00–4.00) | |
| Mean ± SD | 2.94 ± 3.67 | 3.55 ± 4.72 | 2.83 ± 3.48 | |
| From symptom onset to initial COVID-19 diagnosis | | | | |
| Median (interquartile range) | 4.00 (2.00–7.00) | 7.00 (2.00–13.00) | 4.00 (1.75–7.00) | |
| Mean ± SD | 5.23 ± 4.44 | 7.36 ± 5.52 | 4.83 ± 4.14 | |
| From symptom onset to development of pneumonia | | | | |
| Median (interquartile range | 4.00 (2.00–7.50) | 8.00 (2.00–13.00) | 4.00 (2.00–7.00) | |
| Mean ± SD | 5.35 ± 4.46 | 7.73 ± 5.62 | 4.90 ± 4.11 | |
| From development pneumonia to recovery | | | | |
| Median (interquartile range) | 18.00 (16.00–23.00) | 18.00 (12.25–22.50) | 18.00 (16.00–23.00) | |
| Mean ± SD | 18.95 ± 5.49 | 18.10 ± 7.48 | 19.23 ± 4.80 | |
| **Treatments—no. (%)** | | | | |
| Antibiotics | 73 (91.25) | 11 (100.00) | 62 (89.86) | |
| Oseltamivir | 20 (25.00) | 6 (54.55) | 14 (20.29) | |
| Ribavirin, ganciclovir, or peramivir | 47 (58.75) | 3 (27.27) | 44 (63.77) | |
| Umifenovir | 49 (61.25) | 10 (90.91) | 39 (52.17) | |
| Antifungal medications | 10 (12.50) | 0 (0.00) | 10 (14.49) | |
| Systemic glucocorticoids | 29 (36.25) | 0 (0.00) | 29 (42.03) | |
| Nebulized interferon-α inhalation | 70 (87.50) | 10 (90.91) | 60 (86.96) | |
| Lopinavir/ritonavir | 5 (6.25) | 0 (0.00) | 5 (7.25) | |
| Oxygen therapy | 39 (48.75) | 1 (9.09) | 38 (55.07) | |
| High-flow nasal cannula | 11 (13.75) | 0 (0.00) | 11 (15.94) | |
| Mechanical ventilation | | | | |
| Invasive | 2 (2.50) | 0 (0.00) | 2 (2.90) | |
| Non-invasive | 6 (7.50) | 0 (0.00) | 6 (8.70) | |
| Use of extracorporeal membrane oxygenation | 0 (0.00) | 0 (0.00) | 0 (0.00) | |
| Use of continuous renal replacement therapy | 0 (0.00) | 0 (0.00) | 0 (0.00) | |
| Use of intravenous immunoglobulin | 36 (45.00) | 1 (9.09) | 35 (50.72) | |
| **Clinical outcomes** | | | | |
| Intensive care unit admission | 3 (3.75) | 0 (0.00) | 3 (4.35) | |
| Death | 0 (0.00) | 0 (0.00) | 0 (0.00) | |
| Recovery | 47 (58.75) | 10 (90.91) | 37 (53.62) | |
| Hospitalization | 33 (41.25) | 1 (9.09) | 32 (46.38) | |

SD, standard deviation.

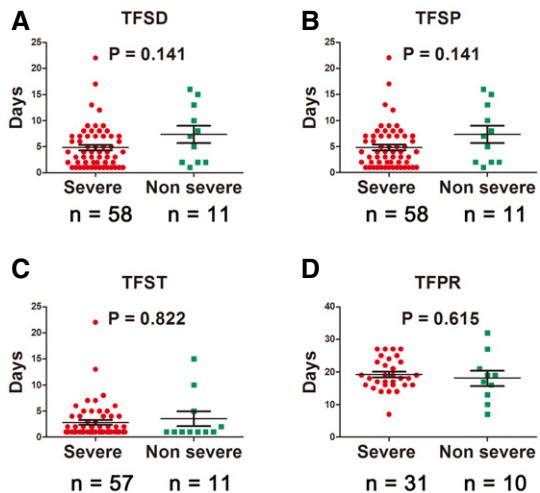

**Figure 2. Time to different type of events in severe versus non-severe COVID-19 patients.**

A   Time from symptom onset to initial diagnosis (TFSD).
B   Time from symptom onset to development of pneumonia (TFSP).
C   Time from symptom onset to treatment (TFST).
D   Time from development of pneumonia to recovery (TFPR).

Data information: Statistical analysis was performed by paired two-tailed Mann–Whitney *U*-test (A–D). Error bars, SEM.
Source data are available online for this figure.

## Discussion

This study showed that most of the severe cases of COVID-19 have the initial symptoms of fever, cough, shortness of breath, and fatigue, while diarrhea, vomiting, and sore throat were not common. Compared with the non-severe cases, the main radiologic findings were bilateral and interstitial abnormalities. In patients with severe disease, more intensive and supportive treatment, including glucocorticoids, human immunoglobulin, interferon (IFN)-α, antibiotics, antiviral therapy (oseltamivir and umifenovir), and oxygen therapy, was administered and relieved symptoms were observed in most cases.

Previous studies have suggested that lymphocytopenia and inflammatory cytokine storm are associated with the severity of infections caused by highly pathogenic coronavirus (de Brito *et al*, 2016; Channappanavar & Perlman, 2017; Gupta *et al*, 2020; preprint: Liu *et al*, 2020b; Yang *et al*, 2020). Cytokines are signaling peptides, proteins, or glycoproteins that are secreted by many cell types, including immune cells, epithelial cells, endothelial cells, and smooth muscle cells. Cytokines allow context-dependent communication within the body (Yiu *et al*, 2012). If the interactions that lead to cytokine production are destabilized, a "cytokine storm" can result, producing unbridled inflammation within tissues and key organs (Yiu *et al*, 2012). Cytokine storms are associated with sepsis and septic shock, influenza, acute respiratory distress, and toxic response to medication and so on (Tanaka *et al*, 2014; Channappanavar & Perlman, 2017; Gupta *et al*, 2020). IL-1, IL-6, IL-10, and TNF-α have been implicated in the 1918 Spanish flu pandemic, the 2003 SARS outbreak and the H5N1 avian influenza infections firstly recognized in 1987 (Channappanavar & Perlman,

2017; Saito *et al*, 2018; Gupta *et al*, 2020). Similarly, recent studies on COVID-19 patients have also reported a decrease in peripheral blood lymphocyte count and an increase in serum inflammatory

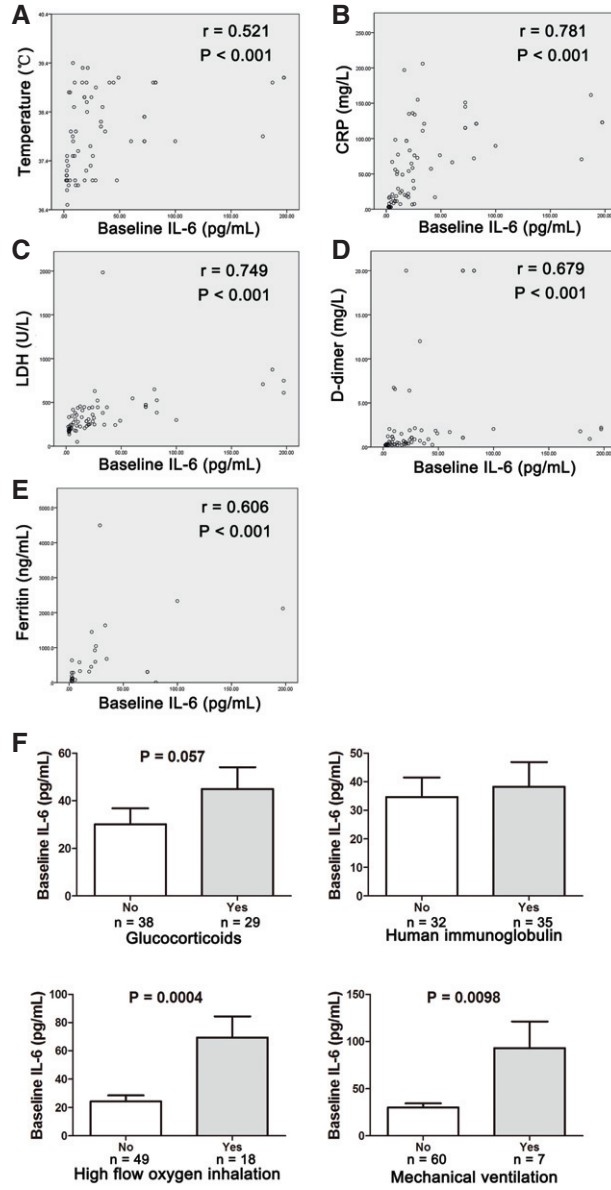

**Figure 3. Correlation between baseline IL-6 level and clinical and laboratory characteristics of severe COVID-19 patients.**

A   Correlation between IL-6 and maximal body temperature during hospitalization.
B   Correlation between IL-6 and C-reactive protein (CRP).
C   Correlation between IL-6 and lactate dehydrogenase (LDH).
D   Correlation between IL-6 and D-dimer.
E   Correlation between IL-6 and ferritin.
F   Levels of baseline IL-6 in patients who received glucocorticoids, human immunoglobulin, high-flow oxygen inhalation, or mechanical ventilation during hospitalization versus patients who did not. Error bars, SEM.

Data information: In (A–E), data were not normally distributed, and correlation was tested by Spearman correlation. In (F), statistical analysis was performed by paired two-tailed Mann–Whitney *U*-test.

cytokine (Huang *et al*, 2020; Liu *et al*, 2020a). Cytokine storms, which can rapidly cause single or multiple organ failure and ultimately can be life-threatening, are considered to be an important cause of death in patients with severe COVID-19. It was recently reported that IL-6, probably derived from inflammatory monocytes, may be responsible for severe lung inflammation and

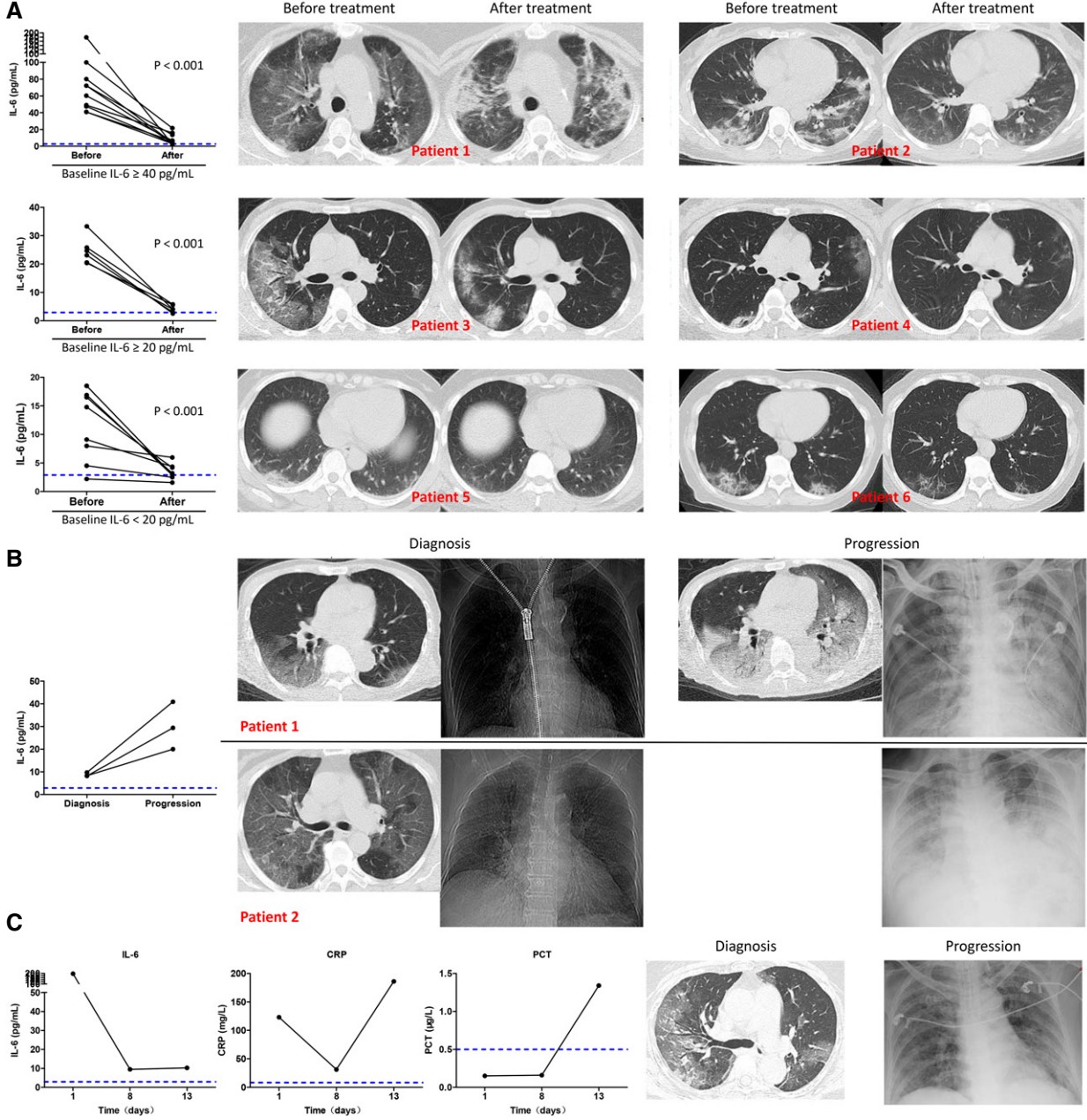

**Figure 4. Variation in IL-6 level and radiological findings in severe COVID-19 patients during disease course.**

A   IL-6 levels before and after treatment in three patient groups classified according to baseline IL-6 levels of ≥ 40 pg/ml (*n* = 11), ≥ 20 pg/ml (*n* = 6), or < 20 pg/ml (*n* = 9; left). Two representative chest computed tomography (CT) scans before and after treatment from each group (right).

B   IL-6 levels at diagnosis and after disease progression in three exacerbated patients (left). Progressed radiological findings were recorded in two patients (right), while radiological assessment after treatment was not performed in the third patient due to poor general condition.

C   The baseline IL-6 level was 197.39 pg/ml in a 69-year-old female patient who showed high fever and dyspnea. IL-6 decreased to 9.47 pg/ml after treatment (day 8), while the symptoms were not relieved. The C-reactive protein (CRP) rebounded and procalcitonin (PCT) increased together with disease exacerbation. Follow-up chest computed tomography (CT) assessment was not performed due to poor general condition, whereas chest X-ray showed aggravated pneumonia. Follow-up sputum culture confirmed the exacerbation was caused by bacterial infection.

Data information: In (A), statistical analysis was performed by paired two-tailed Mann–Whitney U-test. Blue dotted lines denote normal value or normal range.

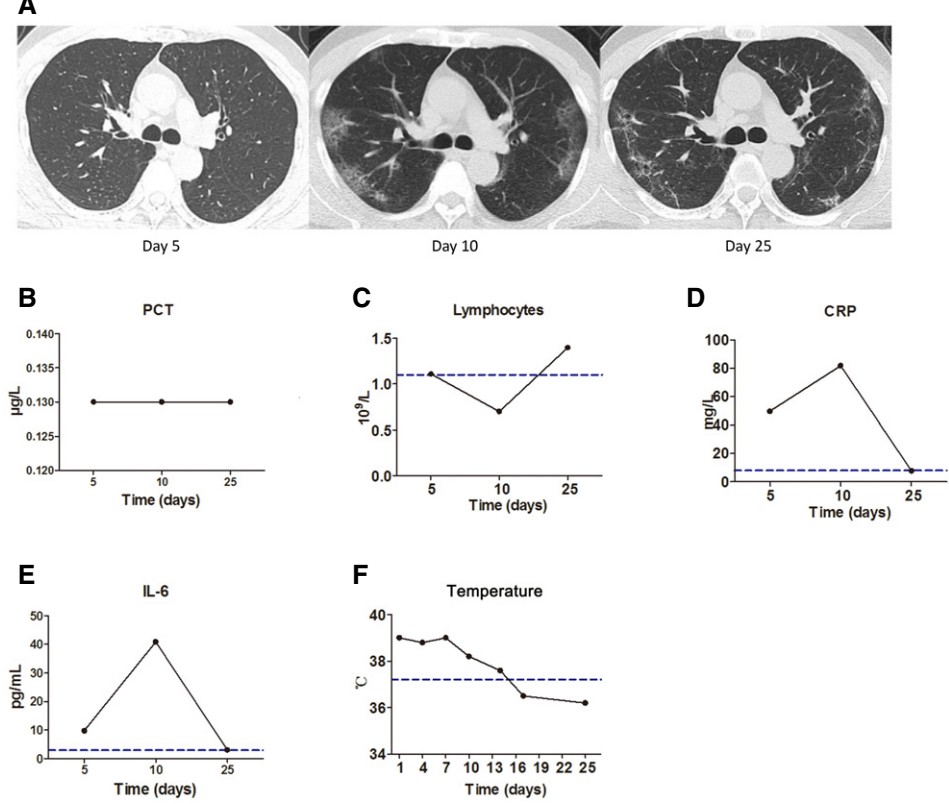

**Figure 5. A case of a 59-year-old male patient diagnosed as severe COVID-19 on the fifth day from the onset of fever.**

A    The chest computed tomography (CT) was normal 5 days after symptom onset. Disease aggravation was evidenced by progressed chest CT images 10 days after symptom onset displaying bilateral multiple patchy ground glass opacities. Disease alleviation was evidenced by improved chest CT images 25 days after symptom onset.

B    Procalcitonin (PCT) levels stayed within normal range throughout the course of disease.

C    Lymphocyte count was normal 5 days after symptom onset, while firstly decreasing and then returning to normal range during the course of disease.

D, E    Both abnormal C-reactive protein (CRP) (D) and IL-6 (E) were detected 5 days after symptom onset when the chest CT and lymphocyte count were still normal. In association with changes in chest CT scans, both CRP and IL-6 further increased 10 days after symptom onset and returned to normal range 25 days after symptom onset.

F    The body temperature was elevated when CRP and IL-6 levels were rising, but returned to normal when CRP and IL-6. Levels were declining.

Data information: Blue dotted lines denote normal value or normal range.

pulmonary function disability in severe COVID-19 patients (preprint: Zhou *et al*, 2020b).

Results of this study indicated that compared with the common type, the changes in immunological parameters and cytokine level in severe type COVID-19 were inconspicuous. There was mild variation in IL-2, IL-4, IL-10, TNF-α, and IFN-γ before and after treatment, all of which fluctuated within the normal range.

Further analysis showed that the changes in IL-6 were associated with severity and disease course of severe COVID-19. Higher baseline IL-6 was associated with more progressed chest CT assessment. Consistently, severe COVID-19 patients who needed more intensive care and treatment, probably due to more severe lung damage, showed higher baseline IL-6 level. In this study, CRP, ferritin, IL-6, and LDH decreased significantly after recovery. In association with disease progression evidenced by exacerbating pulmonary lesions on chest CT scan, IL-6 increased to a further degree. Collectively, our results suggest that IL-6 might be a valuable candidate for monitoring severe type COVID-19. Notably, in one patient with obviously

elevated IL-6 at baseline, IL-6 was markedly decreased, while the clinical presentation was aggravated and disease exacerbation was later proven to be caused by bacterial infection, which may also suggest the potential value of IL-6 in monitoring severe COVID-19.

Interleukin-6 is synthesized by a variety of cells in the lung parenchyma, including alveolar macrophages, type II pneumocytes, T lymphocytes, and lung fibroblasts. IL-6 is a pleiotropic cytokine important in regulating immunological and inflammatory responses (Chen *et al*, 2001; Tanaka *et al*, 2014). IL-6 being an acute phase inflammatory cytokine suggests that measuring circulating IL-6 may reflect the inflammatory state of the lungs (Chen *et al*, 2001; de Brito *et al*, 2016). This is supported by the often observed increase of IL-6 in ARDS and acute complications of lung transplantation (Magnan *et al*, 1996; Maus *et al*, 1998; Wang *et al*, 2004; Tanaka *et al*, 2014). Our findings that IL-6 elevated upon diagnosis and varied correspondingly with disease outcomes support a shared mechanism of cytokine-mediated lung injury caused by viral infection. Siltuximab and tocilizumab are monoclonal antibodies (mAb)

targeted against IL-6 and its receptor (IL6R) (Emery *et al*, 2008; van Rhee *et al*, 2014). Both siltuximab and tocilizumab have been used to treat cytokine release syndrome following chimeric antigen receptor-armed T cells (CART) therapy for leukemia. Since we have found that IL-6 is associated with COVID-19 severity, we suggest that targeting IL-6 may ameliorate cytokine storm-related symptoms in severe COVID-19 cases (Emery *et al*, 2008; Norelli *et al*, 2018). Consistently, promising therapeutic effect of tocilizumab has recently been reported in treating severe COVID-19 patients (Michot *et al*, 2020; Zhang *et al*, 2020).

It is noteworthy to point out that the limitations of our study should not be neglected. Firstly, the tracking of IL-6 variation during disease course had a very limited sample size. Secondly, there were significant differences in the age and gender between severe and non-severe COVID-19 cases enrolled. Thirdly, since this study focused primarily on the severe cases, our findings regarding the potential role of IL-6 in disease monitoring might not be directly applicable to non-severe cases. To address these limitations, studies with expanded sample size and emphasis on non-severe cases of COVID-19 are needed. Lastly, in the correlation study, the IL-6 level was associated with clinical and laboratory parameters indicating a systemic inflammatory response, such as body temperature, CRP, ferritin, and LDH; however, whether IL-6 was the cause of inflammatory response or was a reflection of overall disease status remained to be determined. Our findings have suggested that it is the change in IL-6 level that is of greater value in reflecting and monitoring the evolution of severe COVID-19. Nonetheless, a more detailed recording of IL-6, as well as its upstream and downstream parameters, may help clarify this ambiguity.

In conclusion, severe COVID-19 patients require more intensive treatment, and the prognosis is relatively poor. Elevated IL-6 is associated with disease severity and course, since it decreases with the remission while increases with the aggravation of the disease. Therefore, IL-6 may be a potential marker for disease monitoring in the setting of severe COVID-19. Targeting IL-6 may be effective in treating inflammatory cytokine storm during disease progression. Nonetheless, the precise role of IL-6 in COVID-19 is still not well-studied. A better understanding of IL-6 in the pathogenesis of COVID-19, especially in the severe cases, may help us manage the disease.

# Materials and Methods

### Data source and collection

COVID-19 was diagnosed in accordance with the WHO interim guidance. A confirmed case of COVID-19 was defined as positive for SARS-CoV-2 nucleic acid on high-throughput sequencing or real-time reverse transcriptase polymerase chain reaction (RT–PCR) assays of nasal and pharyngeal swab specimens. Only laboratory-confirmed cases were included in this study, while disease diagnosed based on clinical presentation and radiologic findings, but not on SARS-CoV-2 detection, were excluded. The severity of COVID-19 was classified according to the Guidelines for the Diagnosis and Treatment of Novel Coronavirus (2019-nCoV) Infection (Trial Version 5) issued by the National Health Commission of the People's Republic of China. Severe case was defined when any of

the following criteria was met: (i) dyspnea, respiration rate (RR) $\geq$ 30 times/min; (ii) oxygen saturation by pulse oximeter $\leq$ 93% in resting state; and (iii) partial pressure of arterial oxygen (PaO$_2$) to fraction of inspired oxygen (FiO$_2$) ratio $\leq$ 300 mmHg (l mmHg = 0.133 kPa). We collected data of 69 patients with severe type COVID-19 hospitalized in the Department of Infectious Diseases, Union Hospital, Tongji Medical College, Huazhong University of Science and Technology, between January 21 and February 16, 2020. A retrospective study on the clinical characteristics and laboratory examination was conducted. Eleven non-severe COVID-19 patients were included for comparison. This study was approved by the Ethics Committee of Union Hospital, Tongji Medical College, Huazhong University of Science and Technology and conformed to the principles set out in the WMA Declaration of Helsinki and the Department of Health and Human Services Belmont Report. Verbal consent was obtained from patients before the enrollment. Written informed consent was waived due to the urgent need of data collection.

The clinical symptoms, physical signs, and results of laboratory examination of patients were recorded. Radiologic evaluation included chest CT scan. Laboratory tests included baseline whole blood cell count, blood chemistry, coagulation test, CRP, PCT, LDH, ferritin, ESR, CK, and lymphocyte subset and cytokine profile analysis on admission. In addition, cytokine profile follow-up was conducted in 30 severe type patients after treatment. Treatment plan was recorded. Time from the symptom onset to initial treatment, to initial COVID-19 diagnosis, and to development of pneumonia evidenced by CT scan was recorded, and time from development of pneumonia to discharge was recorded.

### Laboratory diagnosis

Nasal and pharyngeal swab specimens were collected and placed into a collection tube containing preservation solution for virus (Huang *et al*, 2020). Real-time RT–PCR assay for SARS-CoV-2 was conducted by the viral nucleic acid detection kit according to the manufacturer's protocol (Shanghai BioGerm Medical Technology, Co., Ltd.). Laboratory confirmation of COVID-19 was performed by local Center for Disease Control and Prevention (CDC) in accordance with the Chinese CDC protocol. Whenever needed, specimens, including sputum or alveolar lavatory fluid, blood, urine, and feces, were cultured to assess potential bacterial and/or fungal infection that might accompany with SARS-CoV-2 infection.

### Flow cytometry and ELISA detection

The lymphocyte test kit (Beckman Coulter Inc., FL, USA) was used for lymphocyte subset analysis by flow cytometry. Plasma cytokines (IL-2, IL-4, IL-6, IL-10, tumor necrosis factor [TNF]-$\alpha$, and IFN-$\gamma$) were detected by ELISA with human Th1/2 cytokine kit II (BD Ltd., Franklin Lakes, NJ, USA). All tests were performed according to the product manual.

### Statistics

Randomization, blinding, or replication was not applicable in this retrospective study. Continuous variables were described as means

**The paper explained**

**Problem**

The mortality rate of severe coronavirus disease (COVID-19) cases is high. Currently, the monitoring of severe COVID-19 mainly relies on the observation of clinical presentation. Reliable marker(s) for disease monitoring and effective treatment screening are urgently needed.

**Results**

A total of 69 severe COVID-19 cases were included. Baseline interleukin-6 (IL-6) was found significantly increased, which was positively correlated with the maximal body temperature during hospitalization and with increased baseline levels of CRP, LDH, ferritin, and D-dimer. High baseline IL-6 was associated with more progressed chest computed tomography (CT) findings. Significant decrease in IL-6 and improved CT assessment was found in patients during the recovery phase, while IL-6 was further increased in patients with exacerbated disease presentation.

**Impact**

Our results suggest that the dynamic change in IL-6 could be used as a marker for disease monitoring in patients with severe COVID-19.

and standard errors, or medians and interquartile range [IQR] values. Categorical variables were expressed as counts and percentages. Continuous variables that did not conform normal distribution were compared by the two-tailed Mann–Whitney $U$-test. Proportions for categorical variables were compared by the chi-square test and Fisher's exact test as appropriate. Correlations were determined by Spearman rank correlation analysis. All statistical analyses were performed by GraphPad Prism (version 5.0) and SPSS 26.0 (IBM SPSS Statistics 26.0). For all statistical analyses, $P < 0.05$ was considered statistically significant. Respective statistical tests used are stated in the main text and figure legends.

## Data availability

This study includes no data deposited in external repositories.

**Expanded View** for this article is available online.

## Acknowledgements

We thank the patients and their family for their enthusiastic participation in this study. We thank all the doctors, nurses, and staff who are fighting the campaign against COVID-19. We thank Prof. Huan Guo and Dr. Tingting Qin for statistical consultation. This work was supported by the National Natural Science Foundation of China (No. 81602696 to TL), Fundamental Research Funds for the Central Universities of China (No. 2015QN190 to TL), and the National Science and Technology Major Project of China (No. 2018ZX10302204-002-003 to JY).

## Author contributions

TL and JY conceptualized and designed the study, had full access to all data, and took responsibility for data integrity and accuracy of the analysis. JieZ, YY, LZ, HM, and ZL wrote the manuscript. TL, JiaZ, and JC performed the statistical analysis. XZ and YZ prepared the tables. ZX edited the manuscript. LZ, GW, and JY reviewed the manuscript. All authors contributed to data acquisition, analysis, and interpretation, and approved the final version for submission.

## Conflict of interest

The authors declare that they have no conflict of interest.

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
