## [Review Process File · EMBO Molecular Medicine]

The role of interleukin-6 in monitoring severe case of coronavirus disease 2019

Tao Liu, Jieying Zhang, Yuhui Yang, Hong Ma, Zhengyu Li, Jiaoyue Zhang, Ji Cheng, Xiaoyun Zhang, Yanxia Zhao, Zihan Xia, Liling Zhang, Gang Wu, and Jianhua Yi

DOI: [10.15252/emmm.202012421](https://doi.org/10.15252/emmm.202012421)

Corresponding author(s): Jianhua Yi (doctor_yi2017@163.com) , Liling Zhang (lily-1228@hotmail.com), Gang Wu (wuganghustxh@163.com)

Review Timeline:

Submission Date:	31st Mar 20
Editorial Decision:	14th Apr 20
Revision Received:	28th Apr 20
Editorial Decision:	10th May 20
Revision Received:	11th May 20
Accepted:	14th May 20

Editor: Celine Carret

Transaction Report:

14th Apr 2020

Thank you for the submission of your manuscript to EMBO Molecular Medicine. We have now received comments from the two out of the three Reviewers whom we asked to evaluate your manuscript. We have been unable so far, to retrieve the third. Hence, to avoid further delays I am sending the two consistent evaluations of Reviewers 1 and 3 at this time. I will forward Reviewer 2's delayed report, if and as soon as we are able to obtain it. When (within reason) this report does arrive and if it raises additional important issues that have to be addressed to support this study, these would also need to be taken into consideration in your revision. Please note that I would not ask you to consider further-reaching requests with respect to the current evaluations.

As you will see from the comments below, the two referees are enthusiastic about the study and have suggestions and recommendations to further improve conclusiveness and clarity.

We would therefore welcome the submission of a revised version within three to six months (but given the time sensitivity of the topic, the sooner, the better) for further consideration.

Please read below for important editorial formatting that I would need you to consider to accelerate further processing and consult our author's guidelines for proper formatting of your revised article for EMBO Molecular Medicine.

I look forward to receiving your revised manuscript.

*** Instructions to submit your revised manuscript ***

** PLEASE NOTE ** As part of the EMBO Publications transparent editorial process initiative (see our Editorial at <https://www.embopress.org/doi/pdf/10.1002/emmm.201000094>), EMBO Molecular Medicine will publish online a Review Process File to accompany accepted manuscripts.

In the event of acceptance, this file will be published in conjunction with your paper and will include the anonymous referee reports, your point-by-point response and all pertinent correspondence

relating to the manuscript. If you do NOT want this file to be published, please inform the editorial office at contact@embomolmed.org.

To submit your manuscript, please follow this link:

Link Not Available

1) a .doc formatted version of the manuscript text (including Figure legends and tables). Please make sure that the changes are highlighted to be clearly visible to referees and editors alike.

Please make sure to provide

- 5 keywords,
- have updated call outs,
- legends matching figures,
- a conflict of interest statement,
- The paper Explained,
- a statement that informed consent was obtained from all human subjects and that the experiments conformed to the principles set out in the WMA Declaration of Helsinki and the Department of Health and Human Services Belmont Report.
- Author contributions
- in M&M, provide the antibody dilutions that were used for each antibody
- in M&M, the PCR primers used are missing and must be provided
- in M&M, the statistical paragraph should reflect all information that you have filled in the Authors checklist, especially regarding randomization, blinding, replication.
- indicate in legends exact n= and exact p= values, not a range, along with the statistical test used. Some people found that to keep the figures clear, providing an Appendix table Sx with all exact p-values was preferable. You are welcome to do this if you want to.

2) separate figure files*

3) supplemental information as Expanded View and/or Appendix. Please carefully check the authors guidelines for formatting Expanded view and Appendix figures and tables at <https://www.embopress.org/page/journal/17574684/authorguide#expandedview>

4) a letter INCLUDING the reviewers' reports and your detailed responses to their comments (as Word file)

Also, and to save some time should your paper be accepted, please read below for additional information regarding some features of our research articles:

5) The paper explained: EMBO Molecular Medicine articles are accompanied by a summary of the articles to emphasize the major findings in the paper and their medical implications for the non-specialist reader. Please provide a draft summary of your article highlighting

- the medical issue you are addressing,
- the results obtained and

- their clinical impact.

6) Author contributions: the contribution of every author must be detailed in a separate section (before the acknowledgments).

7) EMBO Molecular Medicine now requires a complete author checklist (<https://www.embopress.org/page/journal/17574684/authorguide>) to be submitted with all revised manuscripts.

Please use the checklist as a guideline for the sort of information we need WITHIN the manuscript as well as in the checklist. This is particularly important for animal reporting, antibody dilutions (missing) and exact p-values and n that should be indicated instead of a range.

8) Every published paper now includes a 'Synopsis' to further enhance discoverability. Synopses are displayed on the journal webpage and are freely accessible to all readers. They include a short stand first (maximum of 300 characters, including space) as well as 2-5 one sentence bullet points that summarise the paper. Please write the bullet points to summarise the key NEW findings. They should be designed to be complementary to the abstract - i.e. not repeat the same text. We encourage inclusion of key acronyms and quantitative information (maximum of 30 words / bullet point). Please use the passive voice. Please attach these in a separate file or send them by email, we will incorporate them accordingly.

You are also welcome to suggest a striking image or visual abstract to illustrate your article. If you do please provide a jpeg file 550 px-wide x 400-px high.

9) A Conflict of Interest statement should be provided in the main text

10) As part of the EMBO Publications transparent editorial process initiative (see our Editorial at <http://embomolmed.embopress.org/content/2/9/329>), EMBO Molecular Medicine will publish online a Review Process File (RPF) to accompany accepted manuscripts.

In the event of acceptance, this file will be published in conjunction with your paper and will include the anonymous referee reports, your point-by-point response and all pertinent correspondence relating to the manuscript. Let us know whether you agree with the publication of the RPF and as here, if you want to remove or not any figures from it prior to publication.

11) Please note that we now mandate that all corresponding authors list an ORCID digital identifier. This takes <90 seconds to complete. We encourage all authors to supply an ORCID identifier, which will be linked to their name for unambiguous name identification.

Currently, our records indicate that there is no ORCID associated with your account.

Please click the link below to provide an ORCID:
Link Not Available

12) The system will prompt you to fill in your funding and payment information. This will allow Wiley

to send you a quote for the article processing charge (APC) in case of acceptance. This quote takes into account any reduction or fee waivers that you may be eligible for. Authors do not need to pay any fees before their manuscript is accepted and transferred to our publisher.

Photos 400-800 DPI

*Additional important information regarding figures and illustrations can be found at <http://bit.ly/EMBOPressFigurePreparationGuideline>

***** Reviewer's comments *****

Referee #1 (Remarks for Author):

Overview: The manuscript described 80 SARS-CoV-2-infected cases of COVID-19. This is a pandemic causing devastating social, economic and medical impacts worldwide, so these characterizations and the search for a marker for disease monitoring the severity of in patients is critical and should be published as soon as possible. While larger scale and similar studies have been published already, the specific focus on one cytokine (IL-6) and monitoring fluctuations overtime and with exacerbated illness is relatively novel. There are a few things missing and points to be addressed to strengthen the paper.

A few things that can be addressed:

1. The abstract statements were made without scientific justifications and are overstatements. Please revise.

1) "Patients with severe disease showed obvious lymphocytopenia." Use "significant" (which is justified by a p value), not "obvious"

2) "Baseline interleukin-6 (IL-6) was significantly increased, which was closely related to the maximal body temperature" Please use "positively correlated", not "closely related".

3) "and CT findings," The CT finding is not numerically measurable and is very subjective. Please remove.

4) "Significant decrease in IL-6 and improved CT assessment was found in patients after effective treatment" Please use "during recovery", not "after effective treatment", because currently no treatment is actually shown effective, not to mention the "treatment" in this study is a mixture of variable medications.

5) "The baseline IL-6 was closely related to COVID-19 severity", please use "associated with" not "closely related", as "closely" is an adjective that cannot be statistically or scientifically justified.

6) "and IL-6 was significantly related to the clinical presentation in severe cases" Please remove, as

the previous sentence already said the same thing (redundancy).

7) "Decrease in IL-6 was closely related to treatment effectiveness" Please revise to "Decrease in IL-6 is observed in the recovery phase", as this study cannot justify treatment effectiveness (requires randomized trials to justify "treatment effectiveness")

8) "increase in IL-6 indicated disease exacerbation" Please use "is associated with", not "indicated"

2. The authors should address limitations of the study in a paragraph. First, the tracking of IL-6 with disease progression has a very limited sample size. Second, there are significant differences in age and gender between the severe and nonsevere groups, that the nonsevere cases are significantly younger than the severe cases. Third, this study focuses on the most severe cases and not the more common mild ones, and therefore the findings cannot be applied to patients who present with mild diseases (which is the majority). Fourth, in this correlation study, IL-6 level is associated with clinical parameters suggesting a systemic inflammatory response, such as body temperature, CRP, ferritin and LDH levels. Whether IL-6 is the cause of the inflammatory response or is a reflection of overall disease status remains to be determined.

3. "patients with severe disease showed more common and more prominent level of D-dimer" - what's the P value of D dimer levels in Figure 1? If the P value is not significant, remove D-dimer from the text. If it is significant, label it on the Figure.

4. "Among the enrolled patients with severe disease, 53.62% were cured" - please use recovered, not cured, throughout this manuscript.

5. "The baseline proportion of CD4+ T cells, CD8+ T cells, B cells, natural killer (NK) cells, and the CD4+ T cells/CD8+ T cells ratio are basically within the normal range (Fig 1C)." Please plot out the absolute count instead of proportion, and test for statistical difference. This is because the absolute lymphocyte count is the more biologically and clinical relevant readout, as used in the evaluation of HIV infection (CD4 absolute count <200 defines AIDS, not the proportion of CD4 T cells in the peripheral blood).

6. "The baseline elevated IL-6 was positively correlated with bilateral involvement and interstitial abnormalities (r=0.453, P=0.001)" This is a problematic statement. First, there is no figure representing this statement (the p values in Figure 2 are not statistical significant). Second, how can a continuous numeric variable (IL-6) positively correlate with a binomial yes/no variable (interstitial abnormalities)? Please explain or remove.

7. Please cite Figure 2 in the text. Figure 2 X-axis labels were clipped off.

8. Figure 3 is a major finding but it has major problems.

1) Figure 3A: the P values were labeled wrong. It cannot be 0.000, and it's not the same as the text (0.0001)

2) Figure 3 legend: please do not include result here. Describe A-E as "Correlation of the baseline IL-6 level with CRP (etc)."

3) Please describe correlation test used here. It seems like the authors used Spearman correlation (as the methods stated). However, these are data in normal distribution and should be tested using Pearson correlation (unless statistically tested not normally distributed). Please first use a normality test to test if the data is normally distributed. If yes (which is more likely, as sample size is >30), please use Pearson correlation instead of Spearman correlation (this is for non-parametric or not normally distributed variables). Please remove the linear line in the plot - only regression (R-squared) shows regression line. Correlation results do not use lines.

- 4) Please make the Y axis and X axis labels and numbers bigger, at least 10-12 font size
- 5) Please use the same font in the same figure.

9. "The baseline IL-6 level was closely related to the maximal body temperature during hospitalization ($r=0.521$, $P=0.000$)" The P value cannot be 0.000. This is the limitation of the software used for reporting numbers below 3 digits. Please find out the limit, or at least report as " <0.001 ". "closely related" is not the scientific term. Please use positively correlated.

10. "Meanwhile, it was significantly related to the increase of CRP ($r=0.781$, $P=0.001$), LDH ($r=0.749$, $P=0.001$), ferritin ($r=0.606$, $P=0.001$) and D-dimer ($r=0.679$, $P=0.001$)" Please use "positively correlated", not "significantly related" (this is not a scientific term).

11. "We found tendency showing that, on admission, the lower the IL-6 level, the shorter the time from development of pneumonia to discharge ($r=0.049$, $P=0.763$), whereas the higher the IL-6 level, the shorter the time from symptom onset to development of pneumonia ($r= -0.116$, $P=0.345$)." This statement is wrong. A correlation coefficient r 0.049 or -0.116 means no correlation (it's very close to zero), not to mention the P value is not significant. Please remove this statement.

12. "The elevated level of IL-6 was associated with the administration of glucocorticoids ($r=0.301$, $P=0.001$), human immunoglobulin ($r=0.147$, $P=0.118$), high flow oxygen inhalation ($r=0.251$, $P=0.007$), mechanical ventilation during hospital ($r=0.223$, $P=0.017$)." These are problematic statements. How can a numeric IL-6 level correlate with a binomial yes/no parameter, such as steroid (yes/no), MIG (yes/no), oxygen (yes/no), ventilation (yes/no)? Unless otherwise explained, this statement is wrong and please remove.

13. "SARS-CoV-2 infection can rapidly activate pathogenic T cells to produce granulocyte-macrophage colony stimulating factor (GM-CSF) and IL-6." This sentence is wrong and there is no citation. The cytokine storm associated with SARS is produced by inflammatory macrophages, not T cells. See Stanley Perlman Cell Host Microbe 2016, in which cause-effect relationships were examined. In the preprint Zhou 2020b cited by the authors, the immunological features examined in macrophages and T cells are correlations, not cause-effect. Please remove this statement.

14. All mentions about "are closely related to" need to be corrected to "are associated with". "Closely" is a non-scientific term that is not justifiable by statistical tests and P values.

15. "Collectively, it showed that CRP, IL-6 and LDH were closely related to disease severity." Please use "were associated with", not "closely related to"

16. Page 8, "IL-6 is positively correlated with disease severity" this is wrong. Again, correlation happens only between two numeric variables, and disease severity is not measurable as a number here (while temperature, CRP etc does). Statements on "positively correlated" needs a statistical test and a P value, not a descriptive statement such as "since it decreases with the remission while increases with the aggravation of the disease". Please revise to "Elevated IL-6 is associated with disease severity".

17. "Therefore, IL-6 may be an ideal marker for disease monitoring" use potential, not ideal.

18. Please do not include result interpretation in the figure legend. For example, Figure 1 legend: "B. The procalcitonin (PCT) of patients with severe COVID-19 was basically normal, while the level of erythrocyte sedimentation rate (ESR), ferritin, C-reactive protein (CRP), D-Dimer

and lactate dehydrogenase (LDH) was significantly increased." Please revise it to "The PCT, ESR, ferritin, CRP, D-dimer and LDH levels in severe versus nonsevere COVID19 patients" and vice versa. Please do so for all figure legends.

Another example: Figure 2B "The time from symptom onset to development of pneumonia (TFSP) of patients with severe COVID-19 was shorter, but there was no significant difference ($P \geq 0.05$)." Please remove "was shorter, but there was no significant difference"

19. "Analysis of the epidemiological pattern curve of COVID-19 showed that the overall epidemic pattern was aggregation outbreak." Define "aggregation outbreak" and cite a reference - aggregation outbreak itself is not a scientific term. Use epidemic curve or epidemiological pattern curve (redundant).

20. "Given the rapid spread of COVID-19 and the high mortality rate of severe cases, a better understanding of the clinical features is..." Remove "the rapid spread of COVID-19", because this study which focuses on IL-6 levels does not help triaging on the spread of the virus. It's about disease severity, not spreading. Any study claiming spreading needs to have viral load results.

21. "the storm of inflammatory factors" - use cytokine storm

22. "In this study, by collecting data of severe cases of laboratory-confirmed COVID-19 cases" Please specify collecting what data. Revise into "In this study, we examined the IL-6 cytokine levels with other parameters indicating systemic inflammatory response and clinical severity, such as body temperature, CRP levels, and chest CT findings.

Typos:

1. Hubei Province Province - duplicate Province
2. Please add a space before all brackets: IL6 (pg/ml), not IL6(pg/ml), including text and figures.
3. Please use umifenovir (generic name) instead of abidol (trade name).

Referee #3 (Comments on Novelty/Model System for Author):

This is an observational study on a 69 severe and 11 non-severe COVID19 patients.

Referee #3 (Remarks for Author):

This work is an important observational study comparing clinical markers in severe and non-severe COVID19 patients. Their primary findings are that IL-6, CRP, and LDH correlate with disease severity, and that measuring IL-6 levels is useful for monitoring progression or recovery from disease, as well as effectiveness of therapeutics. They also report one case in which a severe patient had bacterial pneumonia and IL-6 levels were not informative - this could provide valuable insight for care of distinct classes of patients based on clinical microbiology. Overall, this work is a valuable and timely addition to our knowledge of COVID19, and provides additional rationale and potential guidance for ongoing clinical trials of anti-IL-6 treatments in COVID19 patients.

Minor comments:

The abstract should be edited for clarity and removal of redundant statements.

Text in some figures is too small to read.

Additional discussion of what the LDH levels may indicate would be helpful.

Anti-IL-6 therapy has already been tried worldwide with anecdotal reports of promising outcomes. Please reference any case reports that have been published, as well as the other clinical studies that have come out with similar findings regarding IL-6 increases.

The results could be condensed into the short report format.

Replies to reviewers' comments:

We would like to thank the reviewers for their constructive and critical comments, and for pointing out the existing problems in our manuscript. The comments will be responded point by point below:

Reviewer #1 (Remarks for Author)

Overview: The manuscript described 80 SARS-CoV-2-infected cases of COVID-19. This is a pandemic causing devastating social, economic and medical impacts worldwide, so these characterizations and the search for a marker for disease monitoring the severity of in patients is critical and should be published as soon as possible. While larger scale and similar studies have been published already, the specific focus on one cytokine (IL-6) and monitoring fluctuations overtime and with exacerbated illness is relatively novel. There are a few things missing and points to be addressed to strengthen the paper.

A few things that can be addressed:

1. The abstract statements were made without scientific justifications and are overstatements. Please revise.

1) "Patients with severe disease showed obvious lymphocytopenia." Use "significant" (which is justified by a p value), not "obvious"

Answer: Thanks, this has been revised according to your comment.

2) "Baseline interleukin-6 (IL-6) was significantly increased, which was closely related to the maximal body temperature" Please use "positively correlated", not "closely related".

Answer: Thanks, this has been revised according to your comment.

3) "and CT findings," The CT finding is not numerically measurable and is very subjective. Please remove.

Answer: Thank for pointing out the confusion here. We have found that patients with higher baseline IL-6 showed more severe lung damage as evidenced by chest CT scans, as shown in Fig 4A. Therefore, we avoid describing the correlation, but mention the association between high baseline IL-6 and more progressed CT assessments.

4) **"Significant decrease in IL-6 and improved CT assessment was found in patients after effective treatment"** Please use **"during recovery"**, not **"after effective treatment"**, because currently no treatment is actually shown effective, not to mention the **"treatment"** in this study is a mixture of variable medications.

Answer: Thanks, this has been revised according to your comment.

5) **"The baseline IL-6 was closely related to COVID-19 severity"**, please use **"associated with"** not **"closely related"**, as **"closely"** is an adjective that cannot be statistically or scientifically justified.

Answer: Thanks, this has been revised according to your comment.

6) **"and IL-6 was significantly related to the clinical presentation in severe cases"** Please remove, as the previous sentence already said the same thing (redundancy).

Answer: Thanks, this redundant description has been removed.

7) **"Decrease in IL-6 was closely related to treatment effectiveness"** Please revise to **"Decrease in IL-6 is observed in the recovery phase"**, as this study cannot justify treatment effectiveness (requires randomized trials to justify **"treatment effectiveness"**)

Answer: Thanks, we found that this statement was redundant and this inaccuracy has been revised.

8) **"increase in IL-6 indicated disease exacerbation"** Please use **"is associated with"**, not **"indicated"**

Answer: Thanks, this inaccuracy has been revised.

Original abstract:

Progression to severe disease is a difficult problem in treatment coronavirus disease 2019 (COVID-19) patients. The purpose of this study is to explore changes in markers in severe COVID-19 patients during disease course. 69 severe COVID-19 patients were included. Patients with severe disease showed obvious lymphocytopenia. Elevated level of lactate dehydrogenase (LDH), C-reactive protein (CRP), ferritin and D-dimer was found in most severe cases. Baseline interleukin-6 (IL-6) was significantly increased, which was closely related to the maximal body temperature during hospitalization and CT findings, and to the increased baseline CRP, LDH, ferritin and D-dimer. Significant decrease in IL-6 and improved CT assessment was found in patients after effective treatment, while IL-6 was further increased in exacerbated patients. The baseline IL-6 was closely related to COVID-19 severity, and IL-6 was significantly related to the clinical presentation in severe cases. Decrease in IL-6 was closely related to treatment effectiveness, while increase in IL-6 indicated disease exacerbation. Collectively, the dynamic change in IL-6 can be used as a marker for disease monitoring in patients with severe COVID-19.

Revised abstract:

Progression to severe disease is a difficult problem in treating coronavirus disease 2019 (COVID-19). The purpose of this study is to explore changes in markers in severe COVID-19 patients during disease course. 69 severe COVID-19 patients were included. Patients with severe disease showed significant lymphocytopenia. Elevated level of lactate dehydrogenase (LDH), C-reactive protein (CRP), ferritin and D-dimer was found in most severe cases. The baseline IL-6 was associated with COVID-19 severity. Baseline interleukin-6 (IL-6) was significantly increased, which was positively correlated to the maximal body temperature during hospitalization and to the increased baseline CRP, LDH, ferritin and D-dimer. High baseline IL-6 was associated with more progressed chest computed tomography (CT) findings. Significant decrease in IL-6 and improved CT assessment was found in patients during recovery, while IL-6 was further increased in

exacerbated patients. Collectively, our results suggest that the dynamic change in IL-6 can be used as a marker for disease monitoring in patients with severe COVID-19.

2. The authors should address limitations of the study in a paragraph. First, the tracking of IL-6 with disease progression has a very limited sample size. Second, there are significant differences in age and gender between the severe and nonsevere groups, that the nonsevere cases are significantly younger than the severe cases. Third, this study focuses on the most severe cases and not the more common mild ones, and therefore the findings cannot be applied to patients who present with mild diseases (which is the majority). Fourth, in this correlation study, IL-6 level is associated with clinical parameters suggesting a systemic inflammatory response, such as body temperature, CRP, ferritin and LDH levels. Whether IL-6 is the cause of the inflammatory response or is a reflection of overall disease status remains to be determined.

Answer: Thanks for your suggestion. We agree with the proposed limitations and have added an additional paragraph to address them in the discussion.

Revised:

It is noteworthy to point out that the limitations of our study should not be neglected. Firstly, the tracking of IL-6 variation during disease course had a very limited sample size. Secondly, there were significant differences in the age and gender between severe and nonsevere COVID-19 cases enrolled. Thirdly, since this study focused primarily on the severe cases, our findings regarding the potential role of IL-6 in disease monitoring might not be directly applicable to nonsevere cases. To address these limitations, studies with expanded sample size and emphasis on nonsevere cases of COVID-19 are needed. Lastly, in the correlation study, the IL-6 level was associated with clinical and laboratory parameters indicating a systemic inflammatory response, such as body temperature, CRP, ferritin and LDH; however, whether IL-6 was the cause of inflammatory response or was a reflection of overall disease status remained to be determined. Our findings have suggested that it is the change in IL-6 level that is of greater value in reflecting and monitoring the evolution of severe COVID-19. Nonetheless, a more detailed recording of IL-6, as well as its upstream and downstream parameters, may help clarify this ambiguity.

3. "patients with severe disease showed more common and more prominent level of D-dimer" - what's the P value of D dimer levels in Figure 1? If the P value is not significant, remove D-dimer from the text. If it is significant, label it on the Figure.

Answer: Thanks for pointing this mistake out. The P value for D-dimer is <0.001 and it has been added on Fig 1B. Also the demonstration of P value in Figure 1 has been modified.

Revised Figure 1

4. "Among the enrolled patients with severe disease, 53.62% were cured" - please use recovered, not cured, throughout this manuscript.

Answer: Thanks, this inaccuracy has been revised.

5. "The baseline proportion of CD4⁺ T cells, CD8⁺ T cells, B cells, natural killer (NK) cells, and the CD4⁺ T cells/CD8⁺ T cells ratio are basically within the normal range (Fig 1C)." Please plot out the absolute count instead of proportion, and test for statistical difference. This is because the absolute lymphocyte count is the more biologically and clinical relevant readout, as used in the evaluation of HIV infection (CD4 absolute count <200 defines AIDS, not the proportion of CD4 T cells in the peripheral blood).

Answer: Thanks for your suggestion. We agree that reporting the absolute count of T cells, B cells and NK cells would be more meaningful. Unfortunately, the reports in our institution provide only proportion, but not absolute number, as the readout. It is a pity that we could not improve this part of results as your suggestion.

6. "The baseline elevated IL-6 was positively correlated with bilateral involvement and interstitial abnormalities (r=0.453, P=0.001)" This is a problematic statement. First, there is no figure representing this statement (the p values in Figure 2 are not statistical significant). Second, how can a continuous numeric variable (IL-6) positively correlate with a binomial yes/no variable (interstitial abnormalities)? Please explain or remove.

Answer: Thanks for pointing this statistical inaccuracy out. We agree that IL-6 cannot correlate with binomial variable, and we have removed this statement.

However, as shown in Figure 4A, we did observe that patients with higher baseline IL-6 showed more severe lung damage evidenced by CT scans, which ranged from local patchy showing, to bilateral patchy shadowing and to interstitial abnormalities. Therefore, we have added this description in the results.

Removed:

was positively correlated with bilateral involvement and interstitial abnormalities (r=0.453,

P=0.001)

Added:

For the 30 patients whose IL-6 was assessed before and after treatment, 26 patients, classified into three groups according to the baseline IL-6 level of ≥ 40 pg/mL, ≥ 20 pg/mL or < 20 pg/mL, showed significantly reduced IL-6 after treatment (Fig 4A, left). High baseline IL-6 was associated with more severe pulmonary damage as evidenced by chest CT findings, and the decrease in IL-6 after treatment was accompanied with improved CT assessments after treatment (Fig 4A, right).

7. Please cite Figure 2 in the text. Figure 2 X-axis labels were clipped off.

Answer: Thanks for pointing out this error. It has been corrected.

8. Figure 3 is a major finding but it has major problems.

1) Figure 3A: the P values were labeled wrong. It cannot be 0.000, and it's not the same as the text (0.0001)

Answer: Thanks for pointing out this inaccuracy. This is caused by limitation of GraphPad Prism. We have revised the P value to < 0.001 in Fig 3A-E.

2) Figure 3 legend: please do not include result here. Describe A-E as "Correlation of the baseline IL-6 level with CRP (etc)."

Answer: Thanks for the advice. We have removed result and interpretation in the figure legends and the revised figure legends will be shown in the following answer (see below).

3) Please describe correlation test used here. It seems like the authors used Spearman correlation (as the methods stated). However, these are data in normal distribution and should be tested using Pearson correlation (unless statistically tested not normally distributed). Please first use a normality test to test if the data is normally distributed. If yes (which is more likely, as sample size is >30), please use Pearson correlation instead of Spearman correlation (this is for non-parametric or not normally distributed variables). Please remove the linear line in the plot - only regression (R-squared) shows regression line. Correlation results do not use lines.

Answer: Thanks for your suggestion. We have checked our original data again and have confirmed that the IL-6 level was not normally distributed. Therefore, we used Spearman correlation.

The linear lines were removed according to your comment.

4) Please make the Y axis and X axis labels and numbers bigger, at least 10-12 font size

Answer: Thanks for pointing this shortcoming. Modifications have been made.

5) Please use the same font in the same figure.

Answer: Thanks for pointing this shortcoming. Modifications have been made.

Original Figure 3:

Revised Figure 3:

9. "The baseline IL-6 level was closely related to the maximal body temperature during hospitalization ($r=0.521$, $P=0.000$)" The P value cannot be 0.000. This is the limitation of the software used for reporting numbers below 3 digits. Please find out the limit, or at least report as " <0.001 ". "closely related" is not the scientific term. Please use positively correlated.

Answer: Thanks for pointing out this inaccuracy. We have revised the P values to < 0.001 in the text and Figure 3 A-E. The description "closely related" has also been revised according to your comment. (see also in response to comment #10)

10. "Meanwhile, it was significantly related to the increase of CRP ($r=0.781$, $P=0.001$), LDH ($r=0.749$, $P=0.001$), ferritin ($r=0.606$, $P=0.001$) and D-dimer ($r=0.679$, $P=0.001$)" Please use "positively correlated", not "significantly related" (this is not a scientific term).

Answer: Thanks for pointing out this inaccuracy. We have revised the P values to < 0.001 in the text and Figure 3 A-E. The description "significantly related" has also been revised according to your comment.

Original:

The baseline elevated IL-6 was positively correlated with the bilateral involvement and interstitial abnormalities ($r=0.453$, $P=0.001$) and was closely related to the maximal body temperature during hospitalization ($r=0.521$, $P=0.000$) (Fig 3A), Meanwhile, it was significantly related to the increase of CRP ($r=0.781$, $P=0.001$), LDH ($r=0.749$, $P=0.001$), ferritin ($r=0.606$, $P=0.001$) and D-dimer ($r=0.679$, $P=0.001$) (Fig 3B-E).

Revised:

The baseline elevated IL-6 in severe COVID-19 cases was positively correlated to the maximal body temperature during hospitalization ($r = 0.521$, $P < 0.001$) (Fig 3A), Meanwhile, it was positively correlated to the increase of CRP ($r = 0.781$, $P < 0.001$), LDH ($r = 0.749$, $P < 0.001$), ferritin ($r = 0.606$, $P < 0.001$) and D-dimer ($r = 0.679$, $P < 0.001$) (Fig 3B-E).

11. "We found tendency showing that, on admission, the lower the IL-6 level, the shorter the time from development of pneumonia to discharge ($r=0.049$, $P=0.763$), whereas the higher the IL-6 level, the shorter the time from symptom onset to development of pneumonia ($r= -0.116$, $P=0.345$)." This statement is wrong. A correlation coefficient r 0.049 or -0.116 means no correlation (it's very close to zero), not to mention the P value is not significant. Please remove this statement.

Answer: Thanks for pointing out this statistical inaccuracy. As shown in Figure 2, there was no significant difference in the time to different type of events between severe and nonsevere patients. We therefore removed this statement.

Removed:

We found tendency showing that, on admission, the lower the IL-6 level, the shorter the time from development of pneumonia to discharge ($r=0.049$, $P=0.763$), whereas the higher the IL-6 level, the shorter the time from symptom onset to development of pneumonia ($r= -0.116$, $P=0.345$).

12. "The elevated level of IL-6 was associated with the administration of glucocorticoids ($r=0.301$, $P=0.001$), human immunoglobulin ($r=0.147$, $P=0.118$), high flow oxygen inhalation ($r=0.251$, $P=0.007$), mechanical ventilation during hospital ($r=0.223$, $P=0.017$)." These are problematic statements. How can a numeric IL-6 level correlate with a binomial yes/no parameter, such as steroid (yes/no), IVIG (yes/no), oxygen (yes/no), ventilation (yes/no)? Unless otherwise explained, this statement is wrong and please remove.

Answer: Thanks for pointing out this statistical inaccuracy. We agreed that numeric IL-6 level could not correlate with binomial variable, so we removed this statement.

After consulting with the statisticians in our institution, we analyzed the original data in a different way. We found that patients who received high flow oxygen inhalation and mechanical ventilation showed significant higher baseline IL-6 level than those who did not. Patients who received administration of glucocorticoids also showed a tendency of higher baseline IL-6 level, though the

difference was not significant. Namely, patients who required more intensive treatment, probably due to more severe disease, had higher baseline IL-6 level, reflecting the association between IL-6 and disease severity from a different angle. These results have been added in Figure 3F.

Removed:

The elevated level of IL-6 was associated with the administration of glucocorticoids ($r=0.301$, $P=0.001$), human immunoglobulin ($r=0.147$, $P=0.118$), high flow oxygen inhalation ($r=0.251$, $P=0.007$), mechanical ventilation during hospital ($r=0.223$, $P=0.017$).

Revised:

Severe COVID-19 patients who received high flow oxygen inhalation and mechanical ventilation during hospitalization showed significant higher baseline IL-6 level than those who did not (Fig 3F). Patients who received administration of glucocorticoids also showed a tendency of higher baseline IL-6 level, though the difference was not significant.

Added Figure 3F:

13. "SARS-CoV-2 infection can rapidly activate pathogenic T cells to produce granulocyte-macrophage colony stimulating factor (GM-CSF) and IL-6." This sentence is wrong and there is no citation. The cytokine storm associated with SARS is produced by inflammatory macrophages, not T cells. See Stanley Perlman Cell Host Microbe 2016, in which cause-effect relationships were examined. In the preprint Zhou 2020b cited by the authors, the immunological features examined in macrophages and T cells are correlations, not cause-effect. Please remove this statement.

Answer: Thanks for your suggestion. After checking the cited article by Zhou et al. carefully, we agreed that only correlation between cytokine profile detected in COVID-19 and inflammatory monocytes could be justified, but it was not cause-effect relation. We have revised this statement.

Original:

SARS-CoV-2 infection can rapidly activate pathogenic T cells to produce granulocyte-macrophage colony stimulating factor (GM-CSF) and IL-6. GM-CSF will further activate CD14+CD16+ inflammatory monocytes and produce more IL-6 and other inflammatory factors, resulting in a cytokine storm that causes severe immune damage to the lungs and other organs (Zhou et al, 2020b).

Revised:

It is recently reported that IL-6, probably derived from inflammatory monocytes, may be responsible for severe lung inflammation and pulmonary function disability in severe COVID-19 patients (Zhou et al, 2020b).

14. All mentions about "are closely related to" need to be corrected to "are associated with". "Closely" is a non-scientific term that is not justifiable by statistical tests and P values.

Answer: Thanks for pointing this inaccuracy out. We have run through the manuscript and made the revision whenever needed.

15. "Collectively, it showed that CRP, IL-6 and LDH were closely related to disease severity." Please use "were associated with", not "closely related to"

Answer: Thanks, this has been revised according to your comment.

16. Page 8, "IL-6 is positively correlated with disease severity" this is wrong. Again, correlation happens only between two numeric variables, and disease severity is not measurable as a number here (while temperature, CRP etc does). Statements on "positively correlated" needs a statistical test and a P value, not a descriptive statement such as "since it decreases with the remission while increases with the aggravation of the disease". Please revise to "Elevated IL-6 is associated with disease severity".

Answer: Thanks for pointing out this inaccuracy. Revision has been made according to your comment.

Original:

IL-6 is positively correlated with disease severity, since it decreases with the remission while increases with the aggravation of the disease.

Revised:

Elevated IL-6 is associated with disease severity and course, since it decreases with the remission while increases with the aggravation of the disease.

17. "Therefore, IL-6 may be an ideal marker for disease monitoring" use potential, not ideal.

Answer: Thanks, this has been revised according to your comment.

18. Please do not include result interpretation in the figure legend. For example, Figure 1 legend: "B. The procalcitonin (PCT) of patients with severe COVID-19 was basically normal, while the level of erythrocyte sedimentation rate (ESR), ferritin, C-reactive protein (CRP), D-Dimer and lactate dehydrogenase (LDH) was significantly increased." Please revise it to "The PCT, ESR, ferritin, CRP, D-dimer and LDH levels in severe versus nonsevere COVID19 patients" and vice versa. Please do so for all figure legends.

Another example: Figure 2B "The time from symptom onset to development of pneumonia

(TFSP) of patients with severe COVID-19 was shorter, but there was no significant difference (P>0.05)." Please remove "was shorter, but there was no significant difference"

Answer: Thanks for pointing out these shortcoming. Revisions on figure legend have been made and are listed below.

Original legend of Figure 1:

Figure 1. Laboratory findings in patients with severe COVID-19.

A Neutrophils and creatinine level from severe type COVID-19 patients were in the normal range. Compared with nonsevere cases, lymphocytes decreased while creatine kinase (CK) increased significantly in patients with severe COVID-19.

B The procalcitonin (PCT) of patients with severe COVID-19 was basically normal, while the level of erythrocyte sedimentation rate (ESR), ferritin, C-reactive protein (CRP), D-Dimer and lactate dehydrogenase (LDH) was significantly increased.

C Lymphocyte subgroup analysis showed that the proportion of CD4⁺ T cells, CD8⁺ T cells, B cells, natural kill (NK) cells and CD4⁺ T cells/CD8⁺ T cells ratio were within the normal range, and there was no significant difference between the two groups.

D Cytokine profile analysis showed that compared with nonsevere type, there was no difference in the level of IL-2, IL-4, TNF- α and IFN- γ , while IL-10 increased slightly and IL-6 increased significantly in patients with severe COVID-19.

Revised legend of Figure 1:

Figure 1. Laboratory findings in severe versus nonsevere COVID-19 patients.

- A The neutrophil and lymphocyte count, and the level of creatine kinase (CK) and creatinine. Error bars, SEM.
- B The level of procalcitonin (PCT), C-reactive protein (CRP), erythrocyte sedimentation rate (ESR), ferritin, D-dimer and lactate dehydrogenase (LDH). Error bars, SEM.
- C The proportion of total T cells, CD4⁺ T cells, CD8⁺ T cells, B cells and natural kill (NK) cells. Error bars, SEM.
- D The cytokine profile of COVID-19 patients, including IL-2, IL-4, IL-6, IL-10, TNF- α and IFN- γ . Error bars, SEM.

Original legend of Figure 2:

Figure 2. Compared with nonsevere COVID-19 patients.

A The time from symptom onset to initial diagnosis (TFSD).

B The time from symptom onset to development of pneumonia (TFSP) of patients with severe COVID-19 was shorter, but there was no significant difference (P>0.05).

C There was no difference in the time from symptom onset to treatment (TFST).

D The time from development of pneumonia to recovery (TFPR) of patients with severe COVID-19 (P>0.05).

Revised legend of Figure 2:

Figure 2. Time to different type of events in severe versus nonsevere COVID-19 patients.

- A The time from symptom onset to initial diagnosis (TFSD). Error bars, SEM.
- B The time from symptom onset to development of pneumonia (TFSP). Error bars, SEM.
- C The time from symptom onset to treatment (TFST). Error bars, SEM.
- D The time from development of pneumonia to recovery (TFPR). Error bars, SEM.

Original legend of Figure 3:

Figure 3. Correlation between baseline IL-6 level and clinical features of severe COVID-19 patients.

A The baseline IL-6 level was positively correlated with the maximal body temperature during hospitalization.

B The baseline IL-6 level was positively correlated with C-reactive protein (CRP).

C The baseline IL-6 level was positively correlated with lactate dehydrogenase (LDH). D The baseline IL-6 level was positively correlated with ferritin.

E The baseline IL-6 level was positively correlated with D-dimer.

Revised legend of Figure 3:

Figure 3. Correlation between baseline IL-6 level and clinical and laboratory characteristics of severe COVID-19 patients.

A Correlation between IL-6 and maximal body temperature during hospitalization.

B Correlation between IL-6 and C-reactive protein (CRP).

C Correlation between IL-6 and lactate dehydrogenase (LDH).

D Correlation between IL-6 and ferritin.

E Correlation between IL-6 and D-dimer.

F Baseline IL-6 level in patients who received glucocorticoids, human immunoglobulin, high flow oxygen inhalation and mechanical ventilation during hospitalization and in patients who did not. Error bars, SEM.

Original legend of Figure 4:

Figure 4. Variation in IL-6 and chest computed tomography (CT) in severe COVID-19 patients.

A The elevated baseline IL-6 was correlated with the severity assessed by chest CT scan, and the decrease in IL-6 after effective treatment was positively correlated with the improvement in chest CT images.

B Three patients showed elevated IL-6 after treatment, which is associated with disease exacerbation and progressed CT imaging.

C The baseline IL-6 was 197.39 pg/mL in a 69-year-old female patient who showed high fever and dyspnea. IL-6 decreased to 9.47 pg/mL after treatment, but the symptoms were not relieved. CT scan was not performed due to poor general condition, whereas chest X-ray showed aggravated pneumonia. The C-reactive protein (CRP) rebounded and procalcitonin (PCT) increased significantly. Follow up sputum culture confirmed the exacerbation was caused by bacterial infection.

Revised legend of Figure 4:

Figure 4. Variation in IL-6 level and radiological findings in severe COVID-19 patients during disease course.

A The IL-6 level before and after treatment in three patients groups classified according to the baseline IL-6 level of ≥ 40 pg/mL (n = 11), ≥ 20 pg/mL (n = 6) or < 20 pg/mL (n = 9) (left). Two representative chest computed tomography (CT) scans before and after treatment from each group (right).

B The IL-6 level before and after treatment in three exacerbated patients (left). Progressed radiological findings were recorded in two patients (right), while radiological assessment after treatment was not performed in the third patient due to poor general condition.

C The baseline IL-6 level was 197.39 pg/mL in a 69-year-old female patient who showed high fever and dyspnea. IL-6 decreased to 9.47 pg/mL after treatment, while the symptoms were not relieved. The C-reactive protein (CRP) rebounded and procalcitonin (PCT) increased in accompany with disease exacerbation. Follow-up CT assessment was not performed due to

poor general condition, whereas chest X-ray showed aggravated pneumonia. Follow-up sputum culture confirmed the exacerbation was caused by bacterial infection.

Original legend of Figure 5:

Figure 5. A case of 59-year-old male patient diagnosed as COVID-19 on the fifth day from the onset of fever.

A The chest computed tomography (CT) was still normal five days after symptom onset; and the patient presented with initial disease aggravation evidenced by CT scan ten days after symptom onset displaying bilateral multiple patchy ground glass opacities, and subsequent alleviation evidenced by improved CT images 25 days after symptom onset. B The procalcitonin (PCT) level had stayed in the normal range throughout the course of disease.

C The lymphocyte count was still normal five days after symptom onset. It reached nadir and returned to normal during the disease course.

D-E Similar trend of fluctuation was detected in the level of C-reactive protein (CRP) and interleukin-6 (IL-6). Both abnormal CRP and IL-6 were detected five days after symptom onset when the chest CT was still normal. In association with changes in CT scans, both CRP and IL-6 peaked ten days after symptom onset and returned to normal 25 days after symptom onset.

F The body temperature correlated with the variation in CRP and IL-6, as it stayed abnormal during the rising phase while returned to normal during the decline phase of CRP and IL-6.

Revised legend of Figure 5:

Figure 5. A case of a 59-year-old male patient diagnosed as severe COVID-19 on the fifth day from the onset of fever.

A The chest computed tomography (CT) was normal five days after symptom onset. Disease aggravation was evidenced by progressed chest CT images ten days after symptom onset displaying bilateral multiple patchy ground glass opacities. Disease alleviation was evidenced by improved chest CT images 25 days after symptom onset.

B The procalcitonin (PCT) level had stayed within the normal range throughout the course of disease.

C The lymphocyte count was normal five days after symptom onset. It firstly decreased and then returned to normal range during the disease course.

D-E Both abnormal C-reactive protein (CRP) and IL-6 were detected five days after symptom onset when the chest CT and lymphocyte count was still normal. In association with changes in chest CT scans, both CRP and IL-6 further increased ten days after symptom onset and returned to normal range 25 days after symptom onset.

F The body temperature stayed abnormal during the rising phase while returned to normal range during the decline phase of CRP and IL-6.

19. "Analysis of the epidemiological pattern curve of COVID-19 showed that the overall epidemic pattern was aggregation outbreak." Define "aggregation outbreak" and cite a reference - aggregation outbreak itself is not a scientific term. Use epidemic curve or epidemiological pattern curve (redundant).

Answer: Thanks for pointing out this inaccuracy. We have removed the original statement. To be more accurate on describing the epidemiological characteristics of COVID-19, we have revised and cited an article by Wu et al. published in *JAMA* in February.

Removed:

Analysis of the epidemiological pattern curve of COVID-19 showed that the overall epidemic pattern was aggregation outbreak.

Revised:

The epidemic curves of COVID-19 reflected what might be a mixed outbreak pattern, with the early cases suggested a continuous common source, potentially zoonotic spillover at Huanan Seafood Wholesale Market in Wuhan City, while the later cases suggested a propagated source as the virus had begun to be transmitted from person to person (Wu et al. 2020)

Wu Z, McGoogan JM (2020) Characteristics of and Important Lessons From the Coronavirus Disease 2019 (COVID-19) Outbreak in China: Summary of a Report of 72314 Cases From the Chinese Center for Disease Control and Prevention. *Jama*

20. "Given the rapid spread of COVID-19 and the high mortality rate of severe cases, a better understanding of the clinical features is..." Remove "the rapid spread of COVID-19", because this study which focuses on IL-6 levels does not help triaging on the spread of the virus. It's about disease severity, not spreading. Any study claiming spreading needs to have viral load results.

Answer: Thanks, this has been revised according to your comment.

Original:

Given the rapid spread of COVID-19 and the high mortality rate of severe cases,

Revised:

Given the high mortality rate of severe COVID-19 cases,

21. "the storm of inflammatory factors" - use cytokine storm

Answer: Thanks, this has been revised according to your comment.

22. "In this study, by collecting data of severe cases of laboratory-confirmed COVID-19 cases" Please specify collecting what data. Revise into "In this study, we examined the IL-6 cytokine levels with other parameters indicating systemic inflammatory response and clinical severity, such as body temperature, CRP levels, and chest CT findings.

Answer: Thanks, the data collected have been specified.

Original:

In this study, by collecting data of severe cases of laboratory-confirmed COVID-19 cases, we analyzed the clinical characteristics and inflammatory markers in patients with severe type COVID-19 in Wuhan City to explore potential markers for disease monitoring.

Revised:

In this study, by examining the IL-6 and parameters relevant to other systemic inflammatory response and clinical severity, such as body temperature, C-reactive protein (CRP) level, erythrocyte sedimentation rate (ESR) and chest computed tomography (CT) findings, we analyzed the clinical characteristics and inflammatory markers in patients with severe type COVID-19 in Wuhan City to explore potential markers for disease monitoring.

Typos:

1. Hubei Province Province - duplicate Province

2. Please add a space before all brackets: IL6 (pg/ml), not IL6(pg/ml), including text and

figures.

3. Please use umifenovir (generic name) instead of abidol (trade name).

Answer: Thanks for pointing out these drawbacks and typographical errors. All have been revised whenever needed.

Reviewer #2

Comments on Novelty/Model System for Author:

As SARS-CoV-2 is rampant around the world, the number of confirmed cases and deaths continues to increase. The authors reported that the changes of IL-6, a cytokine, in patients of distinct levels of disease severity through clinical data of 69 patients. Taking IL-6 as an index to judge the severity of COVID-19 patients is an interesting idea and, if successfully developed, it may serve as an important biomarker for disease prognosis. However, the manuscript has many obvious errors that need to be corrected before acceptance.

1. There are cursory errors in the text throughout the manuscript, such as "Wuhan Province Province", the words are repeated. The authors are encouraged to go through the manuscript with extra care.

Answer: Thanks a lot for pointing out these drawbacks and we apologize for the carelessness. We have run through the manuscript again and corrected several typographical and grammatical errors.

2. The layout of the captions is acceptable. The captions of Fig1C, 2A, and 2B are partially obscured by the pictures below, thus they are not properly displayed.

Answer: Thanks, Figures has been modified according to your comment.

3. The article has some discrepancies in the discussion with their own results. As mentioned in the "Demographic and clinical characteristics" above, "diarrhea, vomiting and sore throat" are not common, but only diarrhea in discussion.

Answer: Thanks, this has been revised according to your comment.

Original:

This study shows that most of the severe cases of COVID-19 have the initial symptoms of fever, cough, shortness of breath and fatigue, while diarrhea is not common.

Revised:

This study showed that most of the severe cases of COVID-19 have the initial symptoms of fever, cough, shortness of breath and fatigue, while diarrhea, vomiting and sore throat was not common.

4. The logic of the article is simple, i.e., to use IL-6 as an indicator to evaluate the disease severity of the COVID-19-infected patients. In the article, the author also mentioned some other indicators such as lymphocytopenia, increased D-dimer, ESR, LDH, CRP and ferritin, but it remains how well these molecules or cytokines serve as a biomarker for COVID-19 disease progression. Discussion on this potential and other beyond their own study may help the reader to apprehend and appreciate their findings.

Answer: Thanks for your suggestion. Firstly, IL-6 may be responsible for the cytokine storm in severe COVID-19 cases and therefore targeting IL-6 may be of therapeutic value, while the D-dimer,

ESR, LDH, CRP and ferritin lack these characteristics, so we would like to put our emphasis only on IL-6. Secondly, these parameters have been discussed elsewhere (Zhu Z, et al. Int J Infect Dis. 2020; Zheng Z, et al. J Infect. 2020; Huang C, et al. Lancet. 2020), and according to our data, they were not as sensitive as IL-6 in indicating COVID-19 severity, so we did not discuss them further in our manuscript.

5. The author mentioned in the discussion: In one patient whose IL-6 level remained low as disease exacerbated, progression of pulmonary lesions was caused by bacterial infection, which might suggest the specificity of IL-6 in COVID-19. This conclusion is not well justified. It is possible that bacterial infection, if contributes to lung damage, can also cause a significant increase in IL-6. The low level of IL-6 in this case is likely to be other reasons.

Answer: Thanks, we agree with your comment that bacterial infection and the consequent lung damage can also cause increase in IL-6. This may explain the finding that the IL-6 of this patient remained above normal value after initial decrease.

At diagnosis, the obvious elevated IL-6 (day 1) and bilateral pulmonary involvement in chest CT scan suggested a severe SARS-CoV-2 infection, while the low procalcitonin level suggested that bacterial infection at diagnosis was less likely.

In the presence of positive bacterial culture, the disease exacerbation was caused by severe bacterial infection, as evidenced by the aggravated clinical presentation, marked increase in procalcitonin and diffuse effusion in chest X-ray. It is therefore reasonable to speculate that the pulmonary damage was worsened at this time point, yet the IL-6, though still remained elevated, was much lower than initial level. So we believe this may indicate the control of SARS-CoV-2 infection, suggesting the potential value of IL-6 in monitoring COVID-19.

Nonetheless, we agreed that the specificity of IL-6 in COVID-19 was not well justified from one single case. Therefore we revised the statement.

Original:

To our interest, in one patient whose IL-6 level remained low as disease exacerbated, progression of pulmonary lesions was caused by bacterial infection, which might suggest the specificity of IL-6 in COVID-19.

Revised:

To our interest, in one patient with obviously elevated IL-6 at baseline, IL-6 was markedly decreased, while the clinical presentation was aggravated and disease exacerbation was later proven to be caused by bacterial infection, which may also suggests the potential value of IL-6 in monitoring severe COVID-19.

Referee #3 (Remarks for Author):

This work is an important observational study comparing clinical markers in severe and non-severe COVID19 patients. Their primary findings are that IL-6, CRP, and LDH correlate with disease severity, and that measuring IL-6 levels is useful for monitoring progression or recovery from disease, as well as effectiveness of therapeutics. They also report one case in which a severe patient had bacterial pneumonia and IL-6 levels were not informative - this could provide valuable insight for care of distinct classes of patients based on clinical microbiology. Overall, this work is a valuable and timely addition to our knowledge of COVID19, and provides additional rationale and potential guidance for ongoing clinical trials of anti-IL-6 treatments in COVID19 patients.

Minor comments:

1. The abstract should be edited for clarity and removal of redundant statements.

Answer: Thanks, we have run through the manuscript carefully again and several corrections have been made.

2. Text in some figures is too small to read.

Answer: Thanks, the figures have been modified.

3. Anti-IL-6 therapy has already been tried worldwide with anecdotal reports of promising outcomes. Please reference any case reports that have been published, as well as the other clinical studies that have come out with similar findings regarding IL-6 increases.

Answer: Thanks for the suggestions. We have found and cited several newly-published study regarding the therapeutic value of targeting IL-6.

Original:

Since we have found that IL-6 is related to COVID-19 severity, we suggest that targeting IL-6 may ameliorate cytokine storm-related symptoms in severe COVID-19 cases (Emery et al, 2008; Norelli et al, 2018).

Revised:

Since we have found that IL-6 is related to COVID-19 severity, we suggest that targeting IL-6 may ameliorate cytokine storm-related symptoms in severe COVID-19 cases (Emery et al, 2008; Norelli et al, 2018). To our interest, promising therapeutic effect of tocilizumab has recently been reported in treating severe COVID-19 patients (Zhang et al, 2020; Michot et al, 2020).

Zhang X, Song K, Tong F, Fei M, Guo H, Lu Z, Wang J, Zheng C (2020) First case of COVID-19 in a patient with multiple myeloma successfully treated with tocilizumab. *Blood Adv* 4: 1307-1310

Michot JM, Albiges L, Chaput N, Saada V, Pommeret F, Griscelli F, Balleyguier C, Besse B, Marabelle A, Netzer F *et al* (2020) Tocilizumab, an anti-IL6 receptor antibody, to treat Covid-19-related respiratory failure: a case report. *Ann Oncol*

4. The results could be condensed into the short report format.

Answer: Thanks for the suggestions. We have tried our best to present our results as concise as possible, while not sacrificing the integrity and comprehensiveness.

10th May 2020

Thank you for the submission of your revised manuscript to EMBO Molecular Medicine and for sending us the ethical approval. We have now received the enclosed reports from the referees that were asked to re-assess it. As you will see the reviewers are supportive and I am pleased to inform you that we will be able to accept your manuscript pending the following final amendments:

1) In the main manuscript file, please do the following:

- correct/answer the track changes suggested by our data editors by working from the attached document
- add up to 5 keywords
- we note 3 corresponding authors on the paper but 4 in our submission system, please confirm
- in M&M, the statistical paragraph should reflect all information that you have filled in the Authors checklist, especially regarding randomisation, blinding, replication.
- indicate in legends exact n= and exact p= values, not a range, along with the statistical test used. Some people found that to keep the figures clear, providing an Appendix table Sx with all exact p-values was preferable. You are welcome to do this if you want to.
- update the ethics approval accordingly

2) Figures

- From Source Data provided for Figure 2C, we see 57 data points not 58 as indicated in the figure for TFST, revise
- Figures should be provided as portrait not landscape and font must be increased

3) Every published paper now includes a 'Synopsis' to further enhance discoverability. Synopses are displayed on the journal webpage and are freely accessible to all readers. They include a short stand first (maximum of 300 characters, including space) as well as 2-5 one sentence bullet points that summarise the paper. Please write the bullet points to summarise the key NEW findings. They should be designed to be complementary to the abstract - i.e. not repeat the same text. We encourage inclusion of key acronyms and quantitative information (maximum of 30 words / bullet point). Please use the passive voice. Please attach these in a separate file or send them by email, we will incorporate them accordingly.

You are also encouraged to suggest a striking image or visual abstract to illustrate your article. If you do please provide a jpeg file 550 px-wide x (250-400)-px high.

4) As part of the EMBO Publications transparent editorial process initiative (see our Editorial at <http://embomolmed.embopress.org/content/2/9/329>), EMBO Molecular Medicine will publish online a Review Process File (RPF) to accompany accepted manuscripts.

In the event of acceptance, this file will be published in conjunction with your paper and will include the anonymous referee reports, your point-by-point response and all pertinent correspondence relating to the manuscript. Let us know whether you agree with the publication of the RPF, including figures.

Please submit your revised manuscript as soon as possible. I look forward to seeing a revised form of your manuscript soon.

*** Instructions to submit your revised manuscript ***

To submit your manuscript, please follow this link:

Link Not Available

- 1) a .doc formatted version of the manuscript text (including Figure legends and tables)
- 2) Separate figure files*
- 3) supplemental information as Expanded View and/or Appendix. Please carefully check the authors guidelines for formatting Expanded view and Appendix figures and tables at <https://www.embopress.org/page/journal/17574684/authorguide#expandedview>
- 4) a letter INCLUDING the reviewer's reports and your detailed responses to their comments (as Word file).
- 5) Every published paper now includes a 'Synopsis' to further enhance discoverability. Synopses are displayed on the journal webpage and are freely accessible to all readers. They include a short

stand first (maximum of 300 characters, including space) as well as 2-5 one sentence bullet points that summarise the paper. Please write the bullet points to summarise the key NEW findings. They should be designed to be complementary to the abstract - i.e. not repeat the same text. We encourage inclusion of key acronyms and quantitative information (maximum of 30 words / bullet point). Please use the passive voice. Please attach these in a separate file or send them by email, we will incorporate them accordingly.

You are also welcome to suggest a striking image or visual abstract to illustrate your article. If you do please provide a jpeg file 550 px-wide x 400-px high.

***Additional important information regarding Figures**

Photos 400-800 DPI

*Additional important information regarding figures and illustrations can be found at <http://bit.ly/EMBOPressFigurePreparationGuideline>

14th May 2020

We are pleased to inform you that your manuscript is accepted for publication and is now being sent to our publisher to be included in the next available issue of EMBO Molecular Medicine.

We would like to remind you that as part of the EMBO Publications transparent editorial process initiative, EMBO Molecular Medicine will publish a Review Process File online to accompany accepted manuscripts. If you do NOT want the file to be published or would like to exclude figures, please immediately inform the editorial office via e-mail.

Please read below for additional IMPORTANT information regarding your article, its publication and the production process.

Congratulations on your interesting work,

*** ** IMPORTANT INFORMATION ** **

SPEED OF PUBLICATION

The journal aims for rapid publication of papers, using using the advance online publication "Early View" to expedite the process: A properly copy-edited and formatted version will be published as "Early View" after the proofs have been corrected. Please help the Editors and publisher avoid delays by providing e-mail address(es), telephone and fax numbers at which author(s) can be contacted.

Should you be planning a Press Release on your article, please get in contact with embomolmed@wiley.com as early as possible, in order to coordinate publication and release dates.

LICENSE AND PAYMENT:

All articles published in EMBO Molecular Medicine are fully open access: immediately and freely available to read, download and share.

EMBO Molecular Medicine charges an article processing charge (APC) to cover the publication costs. You, as the corresponding author for this manuscript, should have already received a quote with the article processing fee separately. Please let us know in case this quote has not been

received.

Once your article is at Wiley for editorial production you will receive an email from Wiley's Author Services system, which will ask you to log in and will present you with the publication license form for completion. Within the same system the publication fee can be paid by credit card, an invoice, pro forma invoice or purchase order can be requested.

Payment of the publication charge and the signed Open Access Agreement form must be received before the article can be published online.

PROOFS

You will receive the proofs by e-mail approximately 2 weeks after all relevant files have been sent to our Production Office. Please return them within 48 hours and if there should be any problems, please contact the production office at embopressproduction@wiley.com.

Please inform us if there is likely to be any difficulty in reaching you at the above address at that time. Failure to meet our deadlines may result in a delay of publication.

All further communications concerning your paper proofs should quote reference number EMM-2020-12421-V3 and be directed to the production office at embopressproduction@wiley.com.

Corresponding Author Name: Liling Zhang¹,Gang Wu,¹Jianhua Yi

Manuscript Number: EMM-2020-12421